# ENHANCING DESCRIPTIVE IMAGE QUALITY ASSESSMENT WITH A LARGE-SCALE MULTI-MODAL DATASET

## ABSTRACT

With the rapid advancement of Vision Language Models (VLMs), VLM-based Image Quality Assessment (IQA) seeks to describe image quality linguistically to align with human expression and capture the multifaceted nature of IQA tasks. However, current methods are still far from practical usage. First, prior works focus narrowly on specific sub-tasks or settings, which do not align with diverse real-world applications. Second, their performance is sub-optimal due to limitations in dataset coverage, scale, and quality. To overcome these challenges, we introduce **E**nhanced **D**escriptive image **Q**uality **A**ssessment (EDQA). Our method includes a multi-functional IQA task paradigm that encompasses both assessment and comparison tasks, brief and detailed responses, full-reference and non-reference scenarios. We introduce a ground-truth-informed dataset construction approach to enhance data quality, and scale up the dataset to 495K under the brief-detail joint framework. Consequently, we construct a comprehensive, large-scale, and high-quality dataset, named EDQA-495K. We also retain image resolution during training to better handle resolution-related quality issues, and estimate a confidence score that is helpful to filter out low-quality responses. Experimental results demonstrate that EDQA significantly outperforms traditional score-based methods, prior VLM-based IQA models, and proprietary GPT-4V in distortion identification, instant rating, and reasoning tasks. Our advantages are further confirmed by real-world applications including assessing the web-downloaded images and ranking model-processed images. Datasets and codes will be released publicly.

## 1 INTRODUCTION

Image Quality Assessment (IQA) aims to measure and compare the quality of images, expecting to align with human perception. With the emergence of Vision Language Models (VLMs) (Liu et al., 2023a; OpenAI, 2023; Ye et al., 2023b), VLM-based IQA begins to attract more research interest (Wu et al., 2024a;b;d;e; You et al., 2023). These methods leverage VLMs to describe image quality using language, recognizing that language better mirrors human expression, and captures the multifaceted nature of IQA tasks (You et al., 2023). However, existing VLM-based IQA methods still fall short especially in aspects of *functionality* and *performance*.

*Functionality*. There are various application scenarios of IQA, but existing VLM-based IQA models only support a few of them. For example, one scenario involves assessing a single image downloaded from the web, while another requires comparing multiple images handled by different algorithms. Also, image restoration needs to assess an image against a reference, while image generation requests non-reference assessments. Therefore, a superior IQA model should be multi-functional to cater to such diverse scenarios. However, existing methods limit to a specific subset of these tasks, such as single-image assessment (Wu et al., 2024b), multi-image comparison (Wu et al., 2024d), or full-reference setting (You et al., 2023), *etc.* Hence, the limitations in functionality hinder the wide applications of prior methods.

*Performance*. Many IQA methods perform well on some specific datasets but may generalize poorly to other images with different contents or distortions. For instance, Co-Instruct (Wu et al., 2024d) performs well on TID2013 dataset (Ponomarenko et al., 2015) (85.0%), but drops significantly to 50.7% when testing on BAPPS dataset (Zhang et al., 2018). A more comprehensive comparison on our newly created benchmark is given in Fig. 1, where it shows that previous works (Wu et al., 2024b;d) under-perform even within their defined tasks and settings. One potential cause for this is the limited scope of their training datasets. For example, the added distortion category in Q-Instruct (Wu

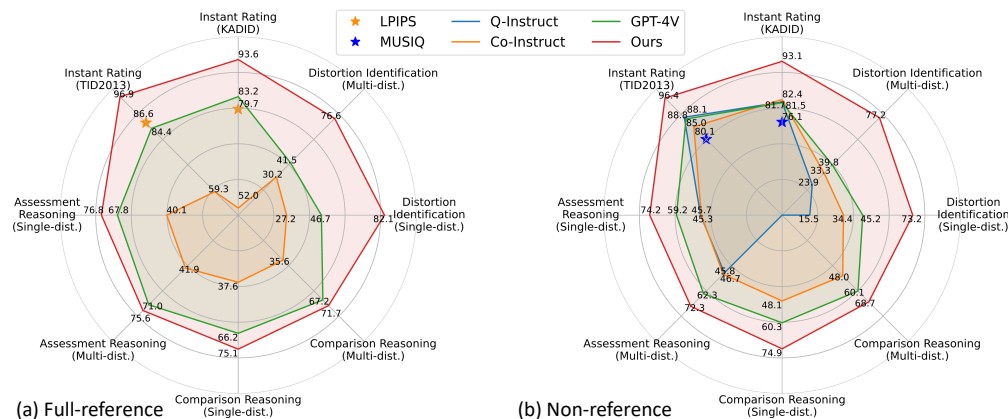

Figure 1: **Performance comparison**. Our model surpasses previous works including Q-Instruct (Wu et al., 2024b), Co-Instruct (Wu et al., 2024d), and the proprietary GPT-4V (OpenAI, 2023) across a broad range of tasks in both full-reference and non-reference settings. Traditional score-based IQA methods like LPIPS (Zhang et al., 2018) and MUSIQ (Ke et al., 2021) have no language abilities, and thus can only be used in *instant rating* task. Q-Instruct is only tested on single-image input tasks.

et al., 2024b) is limited; Co-Instruct (Wu et al., 2024d) directly utilizes GPT-4V (OpenAI, 2023), which is not accurate in IQA tasks, to generate data; and the dataset scale in DepictQA (You et al., 2023) remains small. Additionally, these methods are constrained in their usage by resizing images to a fixed resolution (Wu et al., 2024b;d), while the image resolution is critical for quality assessment. Therefore, the dataset's coverage, quality, and scale together with the training techniques limit the performance of previous methods.

To address these challenges, we propose a multi-functional IQA model to handle various image quality assessment tasks. We categorize these tasks into two types, as shown in Fig. 2. (a) *Single-image assessment* aims to evaluate the quality of a single image by identifying distortions (*e.g.*, "blur" in Fig. 2a top). It can also analyze the distortions' impacts on contents (*e.g.*, blur "affecting the definition of mountains and trees" in Fig. 2a bottom). (b) *Paired-image comparison* focuses on comparing the quality of two distorted images based on the clarity, colorfulness, and sharpness of presented contents. For example, in Fig. 2b, despite reduced contrast, "Image A maintains more scene integrity", as "Image B's serious noise level is more detrimental". We omit multi-image comparison since it is an easy extension of a pairwise one (Gu et al., 2020). Each type includes basic *brief* sub-tasks for fundamental assessments and *detailed* sub-tasks to enhance reasoning abilities. Moreover, the model supports both *full-reference* and non-reference settings, making it adaptable to diverse scenarios.

Under the multi-functional task paradigm, we construct a new large-scale dataset, EDQA-495K, for comprehensive and accurate training and evaluation. First, for diverse distortion, we design and implement 35 types of distortions, each with 5 levels. Second, to enhance the label quality, we inform GPT-4V of the low-level ground truths (*e.g.*, distortions) to leverage its strong high-level perception and language abilities, while avoiding its sub-optimal IQA capabilities. Third, to increase the dataset scale, we scale up the data amount to 495K under the brief-detail combined framework (You et al., 2023). Moreover, our dataset is suitable for both full-reference and non-reference settings.

With EDQA-495K dataset, we then train a VLM model, named **E**nhanced **D**escriptive image **Q**uality **A**ssessment (EDQA). During training, the original image resolution is retained, leading to a better quality perception regarding resolution. Furthermore, we estimate the confidence of responses from key tokens, providing vital auxiliary information, especially for filtering low-quality responses.

The performance of EDQA is evaluated in Fig. 1 and Sec. 5. In brief tasks, our model surpasses general VLMs, IQA-specific VLMs, and score-based IQA methods by a large margin. For example, we achieve 95.9% in non-reference comparison on TID2013 dataset, remarkably surpassing Co-Instruct (85.0%) and GPT-4V (88.1%). In detailed tasks, our model also excels, *e.g.*, recording 74.9% in non-reference comparison reasoning, compared to 48.1% for Co-Instruct and 60.3% for GPT-4V. At last, experiments on real-world applications including assessing web-downloaded images and comparing model-restored images further demonstrate our superiority. We hope that our multi-functional model could serve as a stepping stone towards a unified VLM-based IQA model. Although not yet fully realized, our method showcases the potential of VLM-based IQA models.

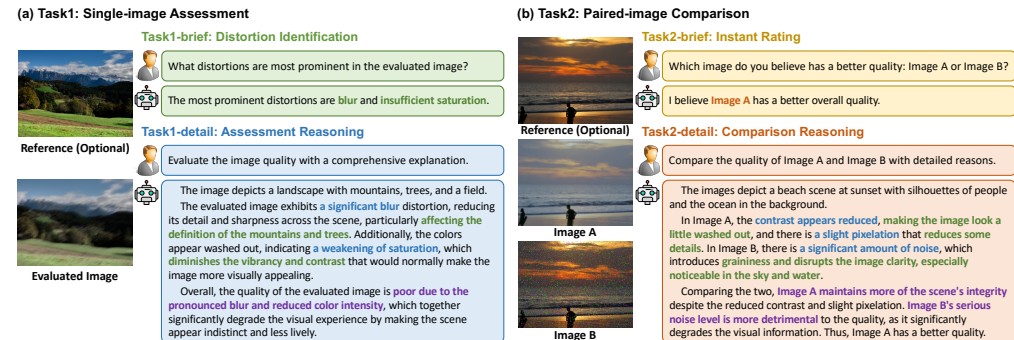

Figure 2: **Task paradigm and qualitative results**. EDQA focuses on two tasks including *single-image assessment* and *paired-image comparison* in both *full-reference* and *non-reference* settings. Each task contains a *brief* sub-task focusing on the fundamental IQA ability, and a *detailed* sub-task fostering the reasoning capacities. More qualitative results in Fig. A10, A11, A12, and A13.

## 2 RELATED WORKS

**Score-based IQA methods**. Traditional IQA methods rely on scores to assess image quality and can be divided into *full-reference* and *non-reference* methods. (a) Full-reference methods compute a similarity score between a distorted image and a high-quality reference. Early works rely on human-designed metrics such as image information (Sheikh and Bovik, 2006), structural similarity (Wang et al., 2004), phase congruency with gradient magnitude (Zhang et al., 2011), *etc.* The rapid advancement of deep learning has also inspired learning-based IQA methods that measure image quality through data-driven training. Pioneered by PieAPP (Prashnani et al., 2018) and LPIPS (Zhang et al., 2018), data-driven approaches (Bosse et al., 2018; Cao et al., 2022; Ding et al., 2020; 2021; Ghildyal and Liu, 2022; Yin et al., 2022; Zhou and Wang, 2022) have spurred innovations in IQA, exhibiting high consistency with human judgments. (b) Non-reference methods directly regress a quality score without a reference image. Initially, human-designed natural image statistics are adopted (Ma et al., 2017; Mittal et al., 2012; 2013; Moorthy and Bovik, 2010; 2011; Saad et al., 2012; Tang et al., 2011). Subsequently, deep-learning-based methods (Kang et al., 2014; Liu et al., 2017; Pan et al., 2018; Su et al., 2020; Sun et al., 2022; Zheng et al., 2021; Zhu et al., 2020) replace hand-crafted statistics by learning quality priors from extensive data. Recent works focus on enhancing performance by introducing multi-scale features (Ke et al., 2021), CLIP pre-training (Wang et al., 2023), multi-dimension attention (Yang et al., 2022), continual learning (Zhang et al., 2022), multitask learning (Zhang et al., 2023b), and so on. However, as discussed in You et al. (2023), score-based IQA methods limit themselves in complex analyses and multi-aspect weighing of IQA, since the information provided by a single score is far from sufficient.

**Vision Language Models** (VLMs) incorporate visual modality into large language models (Chiang et al., 2023; Openai, 2023; Touvron et al., 2023a), aiming to leverage their emergent ability to achieve general visual ability. These VLMs (Alayrac et al., 2022; Dai et al., 2023; Liu et al., 2023a; OpenAI, 2023; Wei et al., 2023; Ye et al., 2023a; Yin et al., 2023; Zhang et al., 2023a; 2024; Zhu et al., 2024a) have demonstrated a general visual ability and can tackle a variety of multi-modality tasks, including image captioning (Agrawal et al., 2019; Chen et al., 2015; Young et al., 2014), visual question answering (Goyal et al., 2017; Liu et al., 2023b; Lu et al., 2022), document understanding (Masry et al., 2022; Mathew et al., 2021; Singh et al., 2019), *etc.* Although proficient in these high-level perception tasks, we demonstrate in Sec. 5 that general-purpose VLMs still struggle with IQA tasks.

**VLM-based IQA methods** aim to achieve better alignment with human perception leveraging the power of VLMs (Wu et al., 2024e). Q-Bench (Wu et al., 2024a) establishes a comprehensive benchmark for evaluating general-purpose VLMs in low-level perception tasks. Zhu et al. (2024b) evaluates various VLMs on the widely-adopted two-alternative forced choice (2AFC) task. Q-Instruct (Wu et al., 2024b) enhances the low-level perception ability of VLMs by introducing a large-scale dataset. Q-Align (Wu et al., 2024c) employs discrete text-defined levels for more accurate quality score regression. Co-Instruct (Wu et al., 2024d) concentrates on the quality comparison among multiple images. DepictQA (You et al., 2023) performs quality description, quality comparison, and comparison reasoning in the full-reference setting. Nonetheless, as highlighted in Sec. 1, these methods focus only on specific aspects of IQA tasks, diverging from the original intents of VLMs' universality and practical usage requirements, and their performance remains sub-optimal.

Table 1: **Overview of our distortion library** with 12 super-categories and 35 sub-categories in total.

| Super-category | Blur | Noise | Compression | Brighten | Darken | Contrast Strengthen | Contrast Weaken | Saturate Strengthen | Saturate Weaken | Over-sharpen | Pixelate | Quantize |
|---|---|---|---|---|---|---|---|---|---|---|---|---|
| # Sub-category | 6 | 6 | 2 | 4 | 4 | 2 | 2 | 2 | 2 | 1 | 1 | 3 |

## 3 TASK PARADIGM AND DATASET CONSTRUCTION

### 3.1 TASK PARADIGM

As highlighted in the introduction, there are various application scenarios for IQA models. First, the evaluation objective can be either single-image assessment or paired-image comparison. The former is useful to rate a web-downloaded image, while the latter suits comparing images processed by two different algorithms. Second, the reference setting may be full-reference or non-reference. For example, image restoration requires assessments based on references, while image generation needs non-reference evaluations. Third, the response could be either brief or detailed. Brief responses suit well-targeted tasks (*e.g.*, comparison without reasons), while detailed responses enhance interpretability and human interaction. To cater to such diverse scenarios, a practical IQA method should be multi-functional. Therefore, we aim to establish such a multi-functional task paradigm for VLM-based IQA research. As shown in Fig. 2, we focus on two tasks, each containing both brief and detailed sub-tasks, and supporting both full-reference and non-reference settings.

- *Task1: single-image assessment*. (a) Brief sub-task: *distortion identification*. Given a distorted image, the model should identify the most obvious distortions. (b) Detailed sub-task: *assessment reasoning*. In addition to identifying distortions, the model should also describe how these distortions affect the perception of image contents and the overall image quality.

- *Task2: paired-image comparison*. (a) Brief sub-task: *instant rating*. Given two distorted images, the model should find the image with better quality. (b) Detailed sub-task: *comparison reasoning*. Building upon the comparison results, the model should first compare the content loss caused by distortions in the two images, then weigh different aspects to draw inferences, and finally justify its comparison results. Note that we omit the multi-image ($>2$) comparison since it can be achieved easily as the extension of paired case (Gu et al., 2020).

Compared with previous works, our design unifies various tasks, response types, and reference settings into a multi-functional paradigm. In contrast, Q-Instruct (Wu et al., 2024b) focuses on non-reference single-image assessment, Co-Instruct (Wu et al., 2024d) targets comparison among multiple images in the non-reference setting, and DepictQA (You et al., 2023) primarily addresses the full-reference setting. Although one can achieve unified IQA by combining these task-specific IQA models, it is impractical due to the significant increase in network parameters, considering that current VLMs are already quite large.

### 3.2 DISTORTION LIBRARY

Existing IQA datasets (*e.g.*, BAPPS (Zhang et al., 2018), PieAPP (Prashnani et al., 2018)) usually introduce distortions (*e.g.*, noise, blur) into high-quality reference images to create distorted images for evaluation. However, these datasets do not publicly release the distortion information of each image, and their distortions only cover limited scenes. Therefore, we aim to develop a comprehensive large-scale distortion library.

**Distortion generation**. Our distortion system comprises 12 super-categories in total, with each super-category consisting of multiple sub-categories. For instance, the "blur" category encompasses "Gaussian blur", "motion blur", "lens blur", *etc.* In total, there are 35 sub-categories. For each sub-category, there are 5 severity levels: "slight", "moderate", "obvious", "serious", and "catastrophic". A summary is illustrated in Tab. 1. Considering the need to assess high-quality images as well, we retain the original image without any distortions in 5% proportion. See details in Appendix B.1.

**Multi-distortion setups**. In practical usage, multiple distortions may occur simultaneously on the same image. While a simple way to simulate them is to add multiple distortions recursively, real-world scenarios are more complex. First, one distortion may weaken another, such as "brighten" weakens "darken", "blur" weakens "over-sharpen". Second, certain distortions exhibit similar visual results, such as "pixelate" looks similar to "blur", making it challenging to identify both if they are

applied simultaneously. We also observe that humans can identify at most two distortions when three or more are applied, as illustrated in Fig. A3. Hence, we limit the distortion number to two and manually review all possible combinations to exclude contradictory or similar combinations. See details of multi-distortion setting in Tab. A1 and Appendix B.1.

## 3.3 DATASET CONSTRUCTION

High-quality and large-scale datasets are crucial for training VLMs. Following Liu et al. (2023a); Yin et al. (2023), training VLMs requires {images, question, response} triplets, where "images" are the ones to be evaluated, "question" describes the task, and "response" is the ground truth answer. In this section, we detail the construction of our dataset from the selection of images and the collection of questions and responses.

**Image collection**. Typical IQA datasets involve two types of images: high-quality reference images and distorted images to be evaluated. Generating distorted images is easy given our comprehensive distortion library introduced in Sec. 3.2. Existing studies often collect a large number of distorted images from a small number of references (Gu et al., 2020; Ponomarenko et al., 2015). However, the semantic richness of images is also crucial for VLM training. Therefore, we primarily source reference images from the KADIS-700K dataset (Lin et al., 2020), which offers 140K pristine reference images from diverse natural and daily scenes. We also leverage other IQA datasets for their convenience to generate responses (details are below).

**Question collection**. Humans often express similar questions using different sentences, necessitating model robustness to various user questions. For each task, we initially prompt GPT-4 (Openai, 2023) to generate 50 candidate questions. Subsequently, we manually eliminate ambiguous and repetitive ones and correct inaccurate ones, creating a question set of 20 questions (see Appendix B.2). These questions are randomly sampled during training and testing to form the data pair.

**Response collection**. We employ two response types as shown in Fig. 3. The first comprises brief templated responses that are easy to produce, where we emphasize the *quantity* to bring robust fundamental skills. The second consists of detailed responses, where we emphasize the *quality* to enhance the model's advanced reasoning abilities. Existing methods to collect detailed responses mainly rely on human annotation (Wu et al., 2024b; You et al., 2023) and GPT-4V generation (Wu et al., 2024d). However, human annotation can be biased and vary in quality particularly when annotators are untrained or tired (You et al., 2023). Also, GPT-4V is not fully reliable since its IQA performance is still unsatisfactory as evidenced in Sec. 5.

We rethink the key aspects of our desired responses and GPT-4V's corresponding abilities, introducing *GT-informed generation* by prompting the Ground Truth (GT) details to enhance GPT-4V's generation. Specifically, a high-quality detailed response should contain image contents, key distortions, the impacts of distortions on contents, and conclusions (*e.g.*, comparison results). While GPT-4V excels at identifying contents and analyzing impacts, it struggles with distortion identification and quality comparison, which will be shown in Sec. 5. To compensate for that, we directly provide it with explicit GT information. The response generation for each task is detailed subsequently.

*Task1-brief: distortion identification*. As shown in Fig. 3a, we first establish a response pool containing 20 templates with unspecified distortions. Next, we add distortions into the reference to create its distorted counterpart and populate a sampled template with the specific distortions to complete the response. For streamlined evaluation, we randomly select half of the questions and append the short answer prompt: "Answer the question using a single word or phrase." Correspondingly, the response will be a single phrase, like "noise", specifying the distortions.

*Task1-detail: assessment reasoning*. Given the reference image, we initially introduce distortions to corrupt the reference. Then, GPT-4V is input with both two images and the distortion information, and requested to assess the quality of the distorted image, as illustrated in Fig. 3b. We instruct GPT-4V to respond from three dimensions: contents, distortions along with their impacts on contents, and overall quality. Here prior studies (Wu et al., 2024b;d) primarily focus on low-level properties, while we consider how these low-level distortions influence the display of high-level contents.

*Task2-brief: instant rating*. We begin by sampling a reference image and its two distorted versions from existing IQA datasets, and then compare the Mean Opinion Score (MOS) to determine the better one, as shown in Fig. 3c. Similar to *distortion identification*, we assemble a response pool

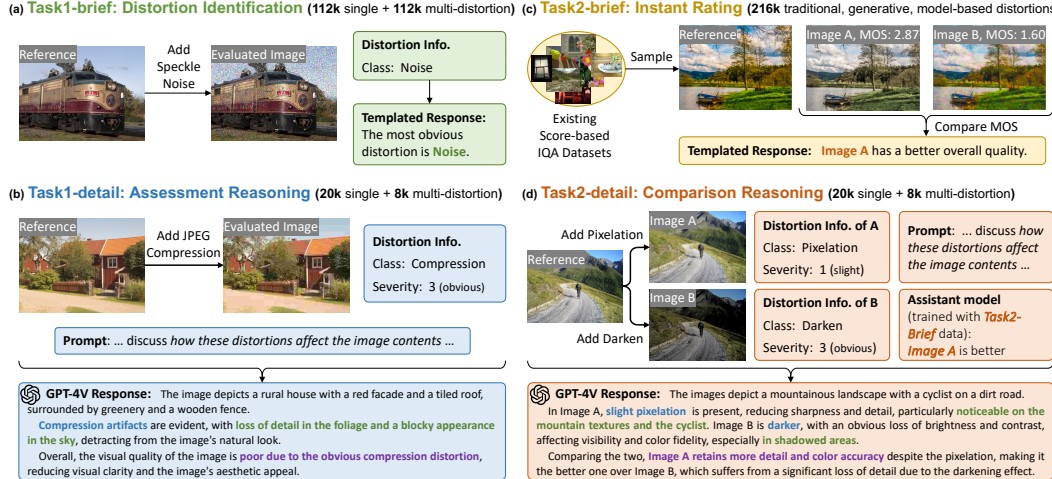

Figure 3: **Construction of EDQA-495K dataset**. For *distortion identification*, templated responses are generated using distortion information. In *instant rating*, we sample images from existing datasets and compare the Mean Opinion Score (MOS) to determine the better image for templated response creation. For *assessment reasoning* and *comparison reasoning* tasks, we provide GPT-4V with evaluated images and Ground Truth (GT) details (*i.e.*, distortion information, comparison results from an assistant model) to facilitate detailed and accurate response generation, called *GT-informed generation*. This additional information is critical as GPT-4V cannot produce it accurately.

of 20 templates to convert the comparison results into textural responses, and append the short answer prompt for the convenience of evaluation. We select three IQA datasets for training, including BAPPS (Zhang et al., 2018), KADID-10K (Lin et al., 2019), and PIPAL (Gu et al., 2020), to cover a diverse range of reference images.

*Task2-detail: comparison reasoning*. As depicted in Fig. 3d, given a high-quality image, we randomly apply distortions to produce two distorted images. We first train an assistant model using the large-scale *instant rating* data to predict the comparison results. Note that GPT-4V does not perform well on the quality comparison task, as shown in our experiments in Tab. 4 and Tab. 8, and thus we train our own comparison model. Then, similar to *assessment reasoning*, we inform GPT-4V of the three images, distortion information, and comparison results to generate detailed responses.

**Setup of non-reference setting**. Our dataset accommodates both full-reference and non-reference settings. However, even for humans, identifying subtle distortions (*e.g.*, minor brightness adjustments) without a reference is challenging. Thus, in the non-reference setting, we selectively remove samples with "slight" severity on some specific distortions, including "brighten", "darken", "contrast weaken", "contrast strengthen", "saturate weaken", "saturate strengthen", "quantize", and "over-sharpen".

**Dataset statistics**. The dataset statistics are illustrated in Tab. 2 (more in Appendix B.2). All tasks, except *instant rating*, are displayed in the single-distortion / multi-distortion format. Our training set contains 439,676 brief samples and 55,577 detailed samples. For *instant rating*, the training set includes BAPPS, KADID, and

Table 2: **Statistics** of our EDQA-495K dataset.

| | Task1-brief *Distortion Identification* | Task1-detail *Assessment Reasoning* | Task2-brief *Instant Rating* | Task2-detail *Comparison Reasoning* |
|---|---|---|---|---|
| Train | 112,000 / 112,000 | 19,829 / 7,981 | 215,676 | 19,809 / 7,958 |
| Validation | 28,000 / 28,000 | 200 / 100 | 41,120 | 200 / 100 |

PIPAL, while the validation set consists of BAPPS, KADID, PIPAL, TID2013 (Ponomarenko et al., 2015), LIVE-MD (Jayaraman et al., 2012), and MDID2013 (Sun et al., 2017). To ensure no intersection between training and validation sets for those overlapped datasets, the original splits are kept. For detailed tasks, all samples in the validation set have been carefully checked by humans.

## 4 MODEL DESIGN

We primarily follow LLaVA-1.6 (Liu et al., 2024) and mPLUG-Owl2 (Ye et al., 2023b) to construct our model, which is composed of a vision encoder, vision abstractor, and LLM. The vision encoder is a frozen CLIP pre-trained ViT-L/14 (Radford et al., 2021). The vision abstractor is a four-layer

transformer, reducing the number of vision tokens to 64 and mapping the vision tokens to textual space. The base LLM is Vicuna-v1.5-7B (Chiang et al., 2023). Following You et al. (2023), we also adopt the unique tag technique, *i.e.*, using specific tags for various types of images, to ensure that the language model can distinguish different input images. To increase robustness, an external high-level perception dataset (detailed description dataset in Yin et al. (2023)) is incorporated during training as a regularization, similar to Wu et al. (2024b); You et al. (2023). See model details in Appendix C. Our experiments in Tab. A9 also show that model architecture has little influence on model performance.

**Retaining resolution in training**. Although previous VLM-based IQA models typically resize all input images to a fixed resolution (Wu et al., 2024b;d), we find this might hurt their performance, as resolution variation may affect visual quality. Instead, we retain the original image resolution during training. Specifically, we interpolate (in bicubic mode) the position embedding in CLIP (Radford et al., 2021) to accommodate varying image resolutions. Ablation studies detailed in Sec. 5.4 demonstrate our model's capability to assess quality variations attributable to resolution, even without explicitly training on such tasks.

**Confidence estimation**. In many applications, it is important to know a confidence score that indicates when the model is uncertain of its response. Here we use the confidence scores of some key tokens as the confidence of the entire answer. Intuitively, the key tokens are distortion names in *distortion identification*, and are either "Image A" or "Image B" in *instant rating*. For detailed reasoning tasks, which feature diverse and non-structured responses, we utilize semantic change testing (Duan et al., 2023) to identify the top 20 tokens with the highest importance scores as key tokens. In semantic change testing, we employ all-MiniLM-L6-v2 (Reimers and Gurevych, 2019) as the similarity model, due to its high processing speed (14K sentences per second). The predicted likelihood of key tokens is averaged as the confidence score. Fig. 6 and Fig. A6 verify that confidence and model performance are highly correlated.

## 5 EXPERIMENTS

### 5.1 METRICS AND BASELINES

**Accuracy, SRCC, and PLCC**. The accuracy metric is utilized for *distortion identification* and *instant rating* tasks. VLMs usually produce diverse textual outputs, and we transform them into brief results for accuracy calculation. Specifically, we prompt our EDQA with "Answer the question using a single word or phrase" to encourage direct output of brief responses. For baseline models, we include all potential answers in the prompt and instruct the model to identify the most accurate one. We emphasize that *our key motivation is to generate descriptive language rather than quality scores*. However, our approach can produce quality scores using pair-wise comparison if required. The quality scores are assessed using Spearman Rank Correlation Coefficient (SRCC) and Pearson Linear Correlation Coefficient (PLCC). The results of quality score are given in Appendix D.1.

**GPT-4 score, BLEU, and ROUGE-L**. We employ the GPT-4 score to evaluate *assessment reasoning* and *comparison reasoning* tasks, following Liu et al. (2023a). Specifically, we provide GPT-4 with both the model-generated response and the corresponding ground truth response. GPT-4 assesses the helpfulness, relevance, accuracy, and level of detail in the model-generated response relative to the ground truth, assigning an overall score on a scale of 0 to 10, where a higher score indicates better quality. This average score is subsequently normalized to a scale of 0 to 100%, reported as the GPT-4 score metric. We further evaluate the reasoning tasks with classical metrics including BLEU and ROUGE-L score following See et al. (2017); Vaswani (2017).

**Baselines**. We categorize our baseline methods into general-purpose VLMs and IQA-specific VLMs. For general VLMs, we include mPLUG-Owl2 (Ye et al., 2023b) (based on LLaMA-2-7B (Touvron et al., 2023b)), LLaVA-1.6 (Liu et al., 2024) (based on Vicuna-v1.5-7B (Chiang et al., 2023)), and the proprietary GPT-4V (OpenAI, 2023). IQA-specific VLMs are represented by Q-Instruct (Wu et al., 2024b), Co-Instruct (Wu et al., 2024d), and CLIP-like LIQE Zhang et al. (2023b). Note that Q-Instruct only supports single-image inputs, thus we only test it on non-reference single-image assessment tasks. Additionally, we compare traditional score-based IQA methods including full-reference ones (PSNR, SSIM (Wang et al., 2004), LPIPS (Zhang et al., 2018), DISTS (Ding et al., 2020)) and non-reference ones (NIQE (Mittal et al., 2013), ClipIQA (Wang et al., 2023), MUSIQ (Ke et al., 2021), MANIQA (Yang et al., 2022)) in *instant rating* task and score regression experiments.

Table 3: **Distortion identification results** under both single-distortion and multi-distortion cases. The accuracy metric is reported in the full-reference / non-reference settings. EDQA greatly outperforms all baselines and maintains its high accuracy in out-of-distribution (OOD) setting.

| | General VLM | | | IQA-specific VLM | | | | |
| | mPLUG-Owl2 | LLaVA-1.6 | GPT-4V | Q-Instruct | Co-Instruct | LIQE | EDQA | EDQA (OOD) |
|---|---|---|---|---|---|---|---|---|
| Single-dist. | 10.1 / 11.6 | 14.0 / 15.3 | 46.7 / 45.2 | - / 15.5 | 27.2 / 34.4 | - / 33.1 | **97.7 / 94.1** | 82.1 / 73.2 |
| Multi-dist. | 10.8 / 10.7 | 12.0 / 12.1 | 41.5 / 39.8 | - / 23.9 | 30.2 / 33.3 | - / 31.4 | **91.3 / 89.3** | 76.6 / 77.2 |

Table 4: **Instant rating results** on multiple benchmarks in the full-reference / non-reference setting with the accuracy metric. Q-Instruct is tested by inputting single images to calculate quality scores, and then compare the scores to rate. EDQA surpasses all baselines by a large margin.

| Methods | | BAPPS$^{test}$ | KADID$^{test}$ | PIPAL$^{test}$ | TID2013 | LIVE-MD | MDID2013 | Mean |
|---|---|---|---|---|---|---|---|---|
| Full-refer. Score-based IQA | PSNR | 68.9 / | 78.7 / | 80.9 / | 85.0 / | 89.7 / | 78.0 / | 80.2 / |
| | SSIM | 69.7 / | 77.1 / | 82.6 / | 78.7 / | 88.1 / | 76.8 / | 78.8 / |
| | LPIPS | 79.4 / | 79.7 / | 84.2 / | 86.6 / | 91.3 / | 85.4 / | 84.4 / |
| | DISTS | 79.7 / | 85.8 / | 84.6 / | 87.0 / | **93.1** / | 88.5 / | 86.5 / |
| Non-refer. Score-based IQA | NIQE | / 49.9 | / 66.9 | / 59.7 | / 65.0 | / 86.9 | / 82.2 | / 68.4 |
| | ClipIQA | / 59.7 | / 75.8 | / 72.6 | / 85.8 | / 65.8 | / 47.0 | / 67.8 |
| | MUSIQ | / 59.2 | / 76.1 | / 77.8 | / 80.1 | / 87.2 | / 81.1 | / 76.9 |
| | MANIQA | / 54.9 | / 68.4 | / 79.2 | / 77.3 | / 75.4 | / 63.5 | / 69.8 |
| General VLM | mPLUG-Owl2 | 50.1 / 50.1 | 50.6 / 50.8 | 49.6 / 49.6 | 48.6 / 48.5 | 49.9 / 50.1 | 50.6 / 50.5 | 49.9 / 49.9 |
| | LLaVA-1.6 | 54.1 / 56.2 | 50.4 / 51.9 | 52.0 / 52.6 | 54.2 / 57.0 | 54.4 / 56.5 | 54.3 / 53.1 | 53.2 / 54.6 |
| | GPT-4V | 70.3 / 63.2 | 83.2 / 81.5 | 78.5 / 78.2 | 84.4 / 88.1 | 79.6 / 72.7 | 70.6 / 67.6 | 77.8 / 75.2 |
| IQA-specific VLM | Q-Instruct | - / 41.6 | - / 81.7 | - / 74.6 | - / 88.8 | - / 73.1 | - / 48.5 | - / 68.1 |
| | Co-Instruct | 49.8 / 50.7 | 52.0 / 82.4 | 50.6 / 72.5 | 59.3 / 85.0 | 50.0 / 70.3 | 50.0 / 58.0 | 52.0 / 69.8 |
| | EDQA (Ours) | **84.7 / 82.4** | **93.6 / 93.1** | **90.5 / 90.0** | **96.9 / 96.4** | **92.1 / 91.8** | **90.0 / 89.6** | **91.3 / 90.6** |

## 5.2 RESULTS ON BENCHMARKS

**Quantitative results of distortion identification** are shown in Tab. 3. First, the performance of Co-Instruct is stably superior in the non-reference setting compared to the full-reference setting, attributed to its training without reference. Second, the performance of open-source general VLMs, including mPLUG-Owl2 and LLaVA-1.6, is still limited, but the proprietary GPT-4V (OpenAI, 2023) outperforms other general-purpose VLMs and exceeds prior specialized IQA VLMs. Third, EDQA significantly surpasses all baseline methods, demonstrating our model's efficacy. Finally, we evaluate our model in an out-of-distribution (OOD) setting. Specifically, for a particular category of distortion (*e.g.*, noise), we use some sub-categories (*e.g.*, Gaussian noise) during training, and different sub-categories (*e.g.*, impulse noise) for evaluation. Results in the last column of Tab. 3 show that our method maintains high accuracy even under such an OOD setting.

**Quantitative results of instant rating** are demonstrated in Tab. 4. First, in the full-reference context, traditional score-based methods, even the simplest PSNR, outperform all general VLMs including GPT-4V and prior IQA-specific VLMs, indicating the inadequacy of existing VLMs in full-reference IQA tasks. Second, conversely, in the non-reference scenario, GPT-4V and Co-Instruct excel beyond most score-based approaches, except MUSIQ. Third, Co-Instruct is trained on multi-image comparison tasks without reference, and thus its performance in full-reference setting drops by quite a large margin. This further demonstrates the necessity of unifying full-reference and non-reference settings. Finally, EDQA demonstrates superior performance across both settings by a large margin, showcasing its substantial advantage.

**Quantitative results of assessment reasoning and comparison reasoning** are shown in Tab. 5 and Tab. 6. First, the performance of the VLM-specific models significantly declines on tasks outside their defined scopes. For instance, Co-Instruct's performance is unsatisfactory on full-reference tasks. Second, GPT-4V shows robust reasoning abilities, stably outperforming prior IQA-specific VLMs. Third, EDQA surpasses GPT-4V, especially in the non-reference setting, affirming its superior reasoning abilities. Finally, EDQA achieves relatively good GPT-4 score and ROUGE-L, indicative of the overall semantic accuracy, but a low BLEU score (yet remains much higher than GPT-4V), which reflects word-level consistency. This suggests that while our predicted answers do not precisely duplicate the ground truths word-for-word, they preserve similar meanings with diverse expressions.

**Qualitative results** of our model on the four tasks in the non-reference setting are depicted in Fig. 2. More qualitative results are provided in Appendix D.4 and Fig. A10, A11, A12, A13, A14.

Table 5: **Assessment reasoning and comparison reasoning results** under both single-distortion and multi-distortion cases. GPT-4 score metric is reported in the full-reference / non-reference setting.

| Methods | Assessment Reasoning | | Comparison Reasoning | |
|---|---|---|---|---|
| | Single-distortion | Multi-distortion | Single-distortion | Multi-distortion |
| GPT-4V | 67.8 / 59.2 | 71.0 / 62.3 | 66.2 / 60.3 | 67.2 / 60.1 |
| Q-Instruct | - / 45.7 | - / 45.8 | - | - |
| Co-Instruct | 40.1 / 45.3 | 41.9 / 46.7 | 37.6 / 48.1 | 35.6 / 48.0 |
| EDQA (Ours) | **76.8 / 74.2** | **75.6 / 72.3** | **75.1 / 74.9** | **71.7 / 68.7** |

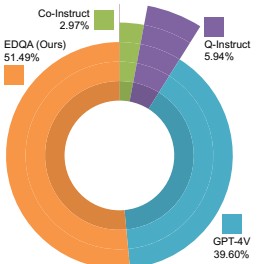

Figure 4: User study.

Table 6: **Assessment reasoning and comparison reasoning results** with classic metrics (BLEU and ROUGE-L) in the full-reference / non-reference setting.

| Methods | Assessment Reasoning | | | | Comparison Reasoning | | | |
|---|---|---|---|---|---|---|---|---|
| | Single-distortion | | Multi-distortion | | Single-distortion | | Multi-distortion | |
| | BLEU | ROUGE-L | BLEU | ROUGE-L | BLEU | ROUGE-L | BLEU | ROUGE-L |
| GPT-4V | 0.020/0.010 | 0.248/0.224 | 0.023/0.015 | 0.246/0.223 | 0.029/0.025 | 0.261/0.243 | 0.030/0.024 | 0.251/0.238 |
| Q-Instruct | - / 0.003 | - / 0.210 | - / 0.002 | - / 0.198 | - | - | - | - |
| Co-Instruct | 0.008/0.005 | 0.201/0.204 | 0.002/0.003 | 0.209/0.203 | 0.039/0.041 | 0.239/0.234 | 0.034/0.036 | 0.239/0.234 |
| EDQA (Ours) | **0.132/0.129** | **0.423/0.422** | **0.180/0.170** | **0.420/0.415** | **0.207/0.207** | **0.466/0.463** | **0.176/0.172** | **0.420/0.413** |

**Quality score regression**. Thought *our key motivation is to generate descriptive language rather than quality scores*, our approach can produce quality scores if required. We transform the score regression problem into instant rating tasks, and calculate the win rate of one image against others (selected by round robin for a small number, random sampling for a large number) as the quality score. The details and results are give in Appendix D.1.

## 5.3 REAL-WORLD APPLICATIONS

**Assessing web-downloaded images**. A practical usage of an IQA model involves assessing the quality of real images. We collect a total of 50 real-world images from the web, featuring diverse contents including animals, plants, faces, buildings, and landscapes. Qualitative results in Fig. 5 and Fig. A14 indicate that our method can assess real images with detailed descriptions. More importantly, EDQA can describe how the distortions affect the contents. For example, in Fig. 5d, our model first accurately identifies the "severe quantization", then describes that the quantization "causes banding in the sky and water", and finally concludes that the quality "is considerably degraded". We also conduct a user study with 20 participants involved. Participants are instructed to choose the assessment result that is of the highest quality among the test methods. The results are shown in Fig. 4, revealing that our approach stably outperforms baseline methods in aligning human perception.

**Comparison on model-processed images**. To develop image restoration models, one often needs to compare the restoration quality of different models. To simulate this, we consider five distortions including "defocus blur", "motion blur", "noise",

Table 7: Results on model-processed images.

| | NIQE | ClipIQA | MUSIQ | ManIQA | GPT-4V | EDQA |
|---|---|---|---|---|---|---|
| Rank ↓ | 2.20 | 1.40 | 1.60 | 1.80 | 1.34±0.27 | **1.20** |
| Accuracy ↑ | 45.5 | 72.7 | 77.3 | 66.4 | 74.5 | **82.7** |

"JPEG compression", and "low resolution". For each distortion, three to four candidate models are used to process the distorted images. We manually rank the restored results, assigning "1" to the best restoration, "2" to the second best, *etc.* Different IQA methods are adopted to compare these restored images pairwise and find the best restoration. The average rank of the found best restoration and the accuracy of the paired comparison are reported in Tab. 7. First, EDQA achieves an average rank of 1.20 (1 is the best), outperforming both GPT-4V and score-based methods. Second, though the temperature is set to 0, GPT-4V shows variability with a large standard deviation. Third, model-restored images are generally out-of-distribution for our model, while EDQA exhibits excellent generalization ability on these images. See details in Tab. A13 and Fig. A8.

## 5.4 ABLATION STUDIES

**Assistant model**. To construct *comparison reasoning* responses, we train an assistant model to predict comparison results (see Fig. 3). These results serve as pseudo labels, which are subsequently

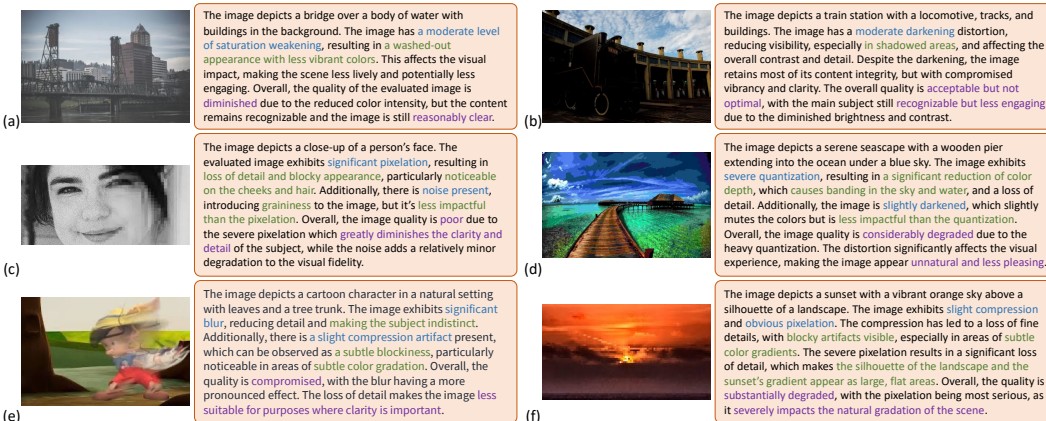

Figure 5: **Qualitative results** on assessing web-downloaded images. More results in Fig. A14.

Table 8: **Our assistant model** surpasses GPT-4V greatly in *instant rating* task. The metric is accuracy in the full-reference / non-reference setting.

| | GPT-4V | Our Assistant |
|---|---|---|
| TID2013 | 84.4 / 88.1 | **94.9 / 94.6** |
| LIVE-MD | 79.6 / 72.7 | **93.1 / 92.8** |
| MDID2013 | 70.6 / 67.6 | **90.1 / 89.8** |

Table 9: **Retaining resolution** is important to identify the images with better aspect ratio or higher resolution.

| Retain Resolution? | | H↔W | 0.8× | 0.9× |
|---|---|---|---|---|
| Training | Inference | | | |
| ✗ | ✗ | 73.0 | 91.7 | 77.2 |
| ✓ | ✗ | 85.6 | 99.0 | 94.8 |
| ✓ | ✓ | **98.8** | **99.3** | **96.8** |

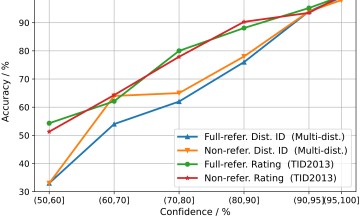

Figure 6: **Confidence** and performance are highly correlated.

provided to GPT-4V to generate responses. We compare the assistant model to GPT-4V on three out-of-distribution IQA datasets. The results in Tab. 8 affirm the superiority of the assistant model.

**Retaining resolution**. In Tab. 9, we study the effects of retaining resolution. We sample 1,000 high-quality images with an aspect ratio greater than 4 : 3. These images are either resized by swapping their height and width (H↔W) or down-sampled by a scale factor of 0.8 or 0.9. The model needs to compare the original and resized images to determine the better one. The alternative of retaining resolution is resizing the two images to a larger resolution, which can maintain the quality difference between the original and resized images (*v.s.*, resizing to a smaller resolution results in two nearly the same images). The results in Tab. 9 prove that retaining resolution is crucial for identifying images with better aspect ratio or higher resolution. More results in Appendix D.2 and Tab. A8.

**Confidence**. We examine the correlation between model performance and estimated confidence in Fig. 6. For *distortion identification* and *instant rating* tasks, across both full-reference and non-reference settings, our model demonstrates improved performance as the confidence interval increases. This validates the effectiveness of our confidence estimation. Details in Appendix D.2 and Fig. A6.

**Model architecture** is studied in Appendix D.2 and Tab. A9, showing little influence on performance.

## 6 CONCLUSIONS AND LIMITATIONS

We introduce EDQA, a VLM-based IQA model, empowered by a new multi-functional task paradigm, dataset enrichment, and training technique, surpassing baseline methods in both benchmarks and two real-world applications, showing the potential of descriptive quality assessment.

**Limitations**. First, the fine-grained abilities requiring more high-level perception skills are still unsatisfactory. For example, in Fig. 5c, though identifying noise and pixelation successfully, our model fails to point out that they are respectively located in the left and right parts. One possible solution is to take the segmentation model to add various distortions to different regions. Second, for the convenience of evaluating, analyzing, and improving the model, we mainly focus on standardized answers. To achieve more flexible responses, LLM rewriting and human annotation can be introduced to increase linguistic diversity during dataset construction. Third, whether our assessment can be used as feedback to improve the quality of generation or restoration models is still under-explored.

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

# APPENDIX

## A    OVERVIEW

This Appendix is structured as follows. Dataset details are described in Appendix B, followed by the details of model, training and inference in Appendix C. More ablation studies, qualitative results, and the details of real-world applications are presented in Appendix D.

## B    DATASET DETAILS

### B.1    DETAILS OF DISTORTION LIBRARY

As stated in Sec. 3.2, to facilitate the dataset construction, we design and implement a comprehensive distortion library. Our distortion system contains 12 distortion super-categories in total, with each category consisting of multiple sub-categories. For instance, the "blur" category encompasses "Gaussian blur", "motion blur", "lens blur", *etc*. In total, there are 35 sub-categories. For each sub-category, there are 5 severity levels: "slight", "moderate", "obvious", "serious", and "catastrophic". In this section, we elaborate on our distortion implementations, including the principles, formulas, and severity setup. We also provide one example for each implementation in Fig. A2, with the reference image in Fig. A1.

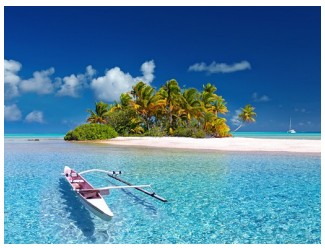

Figure A1: **Reference** for distorted images in Fig. A2.

**Blur**.

- Gaussian blur. The distorted image is generated by convolving the reference image with a Gaussian blur kernel. We set the kernel size $(s_k)$ to be a function of the standard deviation $(\sigma_k)$ of the blur kernel: $s_k = \text{round}(4 \times \sigma_k) + 1$.

- Motion blur. Linear motion blur is applied to the reference image using the linear filter, where $(r, \sigma) \in [(5, 3), (10, 5), (15, 7), (15, 9), (20, 12)]$.

- Glass blur. Filter the image using a Gaussian filter, then randomly jitter each pixel in the image by $x$ pixels, and repeat this process $n$ iterations. $[\sigma, x, n] \in [(0.7, 1, 1), (0.9, 2, 1), (1.2, 2, 2), (1.4, 3, 2), (1.6, 4, 2)]$.

- Lens blur. This distortion uses the circular average filter, where $r \in [1, 2, 4, 6, 8]$.

- Zoom blur. The image is gradually zoomed in and overlaid to calculate the average.

- Jitter blur. Each pixel is randomly displaced by a shift of $\text{randint}(-p, p)$ pixels both in $x$ and $y$ dimensions, with a total of 5 displacements, where $p \in [1, 2, 3, 4, 5]$.

**Noise**.

- Gaussian noise in RGB space. Additive Gaussian noise is applied to each of the RGB channels of an image, where $\sigma \in [0.05, 0.1, 0.15, 0.2, 0.25]$.

- Gaussian noise in YCrCb space. Similar to the Gaussian noise in RGB space, this distortion is implemented in YCbCr space, where $(\sigma_l, \sigma_r, \sigma_b) \in [(0.05, 1, 1), (0.06, 1.45, 1.45), (0.07, 1.9, 1.9), (0.08, 2.35, 2.35), (0.09, 2.8, 2.8)]$.

- Speckle Noise. Speckle Noise is also known as Multiplicative Gaussian noise, where $\sigma \in [0.14, 0.21, 0.28, 0.35, 0.42]$.

- Spatially correlated noise. The reference image is first corrupted by an additive Gaussian noise, which results in each pixel being corrupted by an independent and identically distributed noise pattern. The resultant image is then filtered with an average filter of kernel size $3 \times 3$, correlating the intensity of each pixel with those of the neighboring pixels. More specifically, the distorted image is given by:

$$I_D(x, y, c) = \frac{1}{|N_n|} \sum_{i \in N_n} (I_R(x_i, y_i, c_i) + N(x_i, y_i, c_i)), \tag{A1}$$

where $I_D$ is the distorted image, $I_R$ is the reference image, $N_n$ is the set of neighboring pixels, and $N(x, y, c) \sim \mathcal{N}(0, \sigma_g^2)$.

- Poisson noise. This distortion generates Poisson noise based on the image pixel values, where $intervals \in [80, 60, 40, 25, 15]$.

- Impulse noise. Impulse noise is also known as salt and pepper noise. The density of the noise: $d \in [0.01, 0.03, 0.05, 0.07, 0.10]$.

**Compression**.

- JPEG. The distorted image is a JPEG-compressed version of the reference image, where the parameter in Pillow, quality $q \in [25, 18, 12, 8, 5]$.

- JPEG 2000. This distortion is an advanced compression widely used, where the Pillow's parameter quality $q \in [29, 27.5, 26, 24.5, 23]$.

**Brightness**.

- Brightness shift in HSV space. The RGB image is mapped to HSV, and then we enhance and reduce the brightness by V channel, where $\sigma \in [0.1, 0.2, 0.3, 0.4, 0.5]$ for Brightening and $\sigma \in [-0.1, -0.2, -0.3, -0.4, -0.5]$ for darkening.
- Brightness shift in RGB space. We enhance and reduce the brightness in all channels, where $\sigma \in [0.1, 0.15, 0.2, 0.27, 0.35]$ for Brightening and $\sigma \in [-0.1, -0.15, -0.2, -0.27, -0.35]$ for darkening.
- Gamma Brightness tuning in HSV space. The RGB image is mapped to HSV space and then we enhanced and reduce the brightness by V channel with a gamma function, where $\gamma \in [0.7, 0.58, 0.47, 0.36, 0.25]$ for brightening and $\gamma \in [1.5, 1.8, 2.2, 2.7, 3.5]$ for darkening.

**Contrast**.

- Contrast tuning by scaling. Given an input image $I_{in}$, there is a corresponding $I_{mean}$, which is a gray image in which each element is the mean of $I_{mean}$. The distorted image $I_D$ is generated as following: $I_D = I_{mean} * (1.0 - \alpha) + I_{in} * \alpha$, where $\alpha \in [0.75, 0.6, 0.45, 0.3, 0.2]$ for strengthening and $\alpha \in [1.4, 1.7, 2.1, 2.6, 4.0]$ for weakening.
- Contrast tuning by stretching. Contrast changing is performed as follows: $I_D(x, y, c) = 1/(1 + (\frac{\bar{I}_C}{I_R(x,y,c)+\epsilon})\alpha)$, where $I_D$ is the distorted image, $I_R$ is the reference image, and $\bar{I}_C$ is the mean intensity for channel $c$. $\alpha \in [1.0, 0.9, 0.8, 0.6, 0.4]$ for weakening, and $\alpha \in [2.0, 4.0, 6.0, 8.0, 10.0]$ for strengthening.

**Saturate**.

- Saturate tuning in HSV space. The reference image is firstly mapped into HSV space and then the S channel is scaled, where the scale factor $s \in [0.7, 0.55, 0.4, 0.2, 0.0]$ for weakening and $s \in [3.0, 6.0, 12.0, 20.0, 64.0]$ for enhancement.
- Saturate tuning in YCbCr space. The reference image $I_R$ is firstly mapped into YCbCr space and then the distorted image $I_D$ is generated like the following formulation:

$$I_D(x, y, Cb) = 128 + (I_R(x, y, Cb) - 128) \times s, \tag{A2}$$

$$I_D(x, y, Cr) = 128 + (I_R(x, y, Cr) - 128) \times s, \tag{A3}$$

where $s \in [0.6, 0.4, 0.2, 0.1, 0.0]$ donates the scale factor for weakening and $s \in [2.0, 3.0, 5.0, 8.0, 16.0]$ for strengthening.

**Over-sharpen**. The reference image $I_R$ is firstly processed by a Gaussian blur kernel to generated a blurred image $I_{blur}$. Then the original image is over-sharpened with $\text{cv2.addWeighted}(I_R, 1 + \alpha, I_{blur}, -\alpha, 0)$, where $\alpha \in [2, 2.8, 4, 6, 8]$.

**Pixelate**. The reference image is firstly down-sampled in BOX mode, then up-sampled to the original resolution in NEAREST mode, where the down-sampling factor $\sigma \in [0.5, 0.4, 0.3, 0.25, 0.2]$.

**Quantize**.

- Color quantization using histogram equalization. The color elements are divided into an equal histogram for quantization, where the number of classes $c \in [24, 16, 8, 6, 4]$.
- Color quantization using histogram median. This distortion is implemented by the function PIL.Image.Quantize.MEDIANCUT, where the number of classes $c \in [20, 15, 10, 6, 3]$.
- Color quantization using OTSU method, which is implemented by existing function skimage.filters.threshold_multiotsu to generate thresholds. The number of classes $c \in [15, 11, 8, 5, 3]$.

**Multi-distortion setups**. As discussed in Sec. 3.2, multiple distortions may occur simultaneously on the same image in practical usage. First, we observe that humans can identify at most two distortions when three or more are applied, as in Fig. A3, thus we limit the number of applied distortions to two.

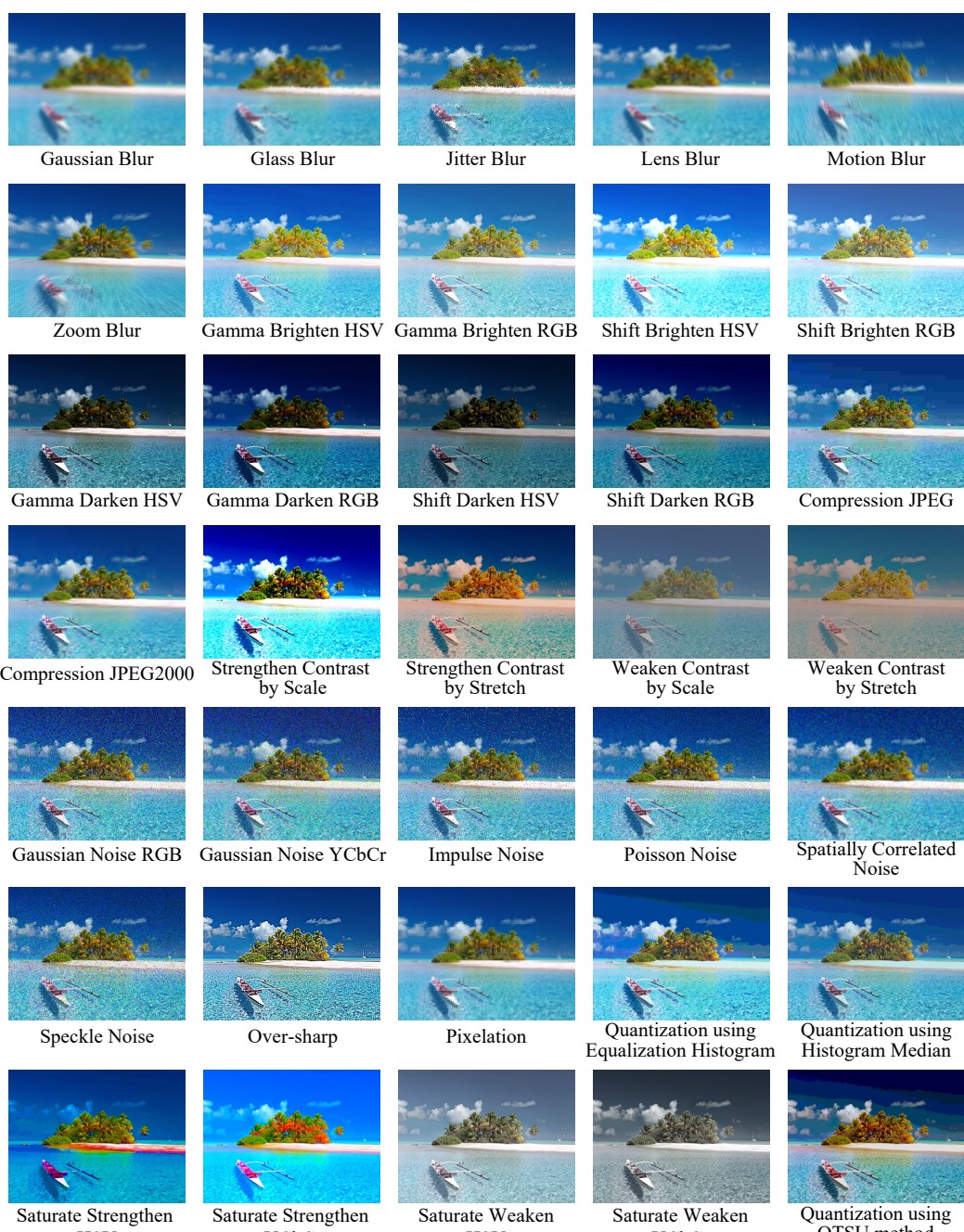

Figure A2: **Distortion examples** of our distortion design. We showcase one example for each distortion implementation. The reference image is depicted in Fig. A1.

Second, some distortions could weaken each other's presentation (*e.g.*, "brighten" weakens "darken", "blur" weakens "over-sharpen"). Also, certain distortions show similar visual effects (*e.g.*, "pixelate" looks similar to "blur"), making it hard to identify both if applied simultaneously. Hence, to exclude contradictory or similar distortion combinations, we manually review all possible combinations. All feasible distortion combinations used in our dataset are provided in Tab. A1.

**Out-of-distribution setups**. In Tab. 3, we evaluate our model in an out-of-distribution (OOD) setting. Specifically, for a particular category of distortion (*e.g.*, noise), we use some sub-categories (*e.g.*, Poisson noise) during training, and different sub-categories (*e.g.*, impulse noise) for evaluation. Here we provide a detailed split of training distortions and evaluation distortions in Tab. A2.

Table A1: **Multi-distortion setting** where we show all feasible distortion combinations.

| First Distortion | All Possible Second Distortions |
|---|---|
| Blur | Brighten, Compression, Contrast Strengthen, Contrast Weaken, Darken, Noise, Quantize, Saturate Strengthen, Saturate Weaken |
| Brighten | Blur, Compression, Noise, Pixelate, Quantize |
| Compression | Blur, Brighten, Contrast Strengthen, Contrast Weaken, Darken, Noise, Saturate Strengthen, Saturate Weaken |
| Contrast Strengthen | Blur, Compression, Noise, Pixelate, Quantize |
| Contrast Weaken | Blur, Compression, Noise, Pixelate, Quantize |
| Darken | Blur, Compression, Noise, Pixelate, Quantize |
| Noise | Blur, Brighten, Compression, Contrast Strengthen, Contrast Weaken, Darken, Over-sharpen, Pixelate, Saturate Strengthen, Saturate Weaken |
| Over-sharpen | Brighten |
| Pixelate | Brighten, Contrast Strengthen, Contrast Weaken, Darken, Noise, Over-sharpen, Quantize, Saturate Strengthen, Saturate Weaken |
| Quantize | Brighten, Contrast Strengthen, Contrast Weaken, Darken, Noise, Over-sharpen, Pixelate, Saturate Strengthen, Saturate Weaken |
| Saturate Strengthen | Blur, Compression, Noise, Over-sharpen, Pixelate, Quantize |
| Saturate Weaken | Blur, Compression, Noise, Over-sharpen, Pixelate, Quantize |

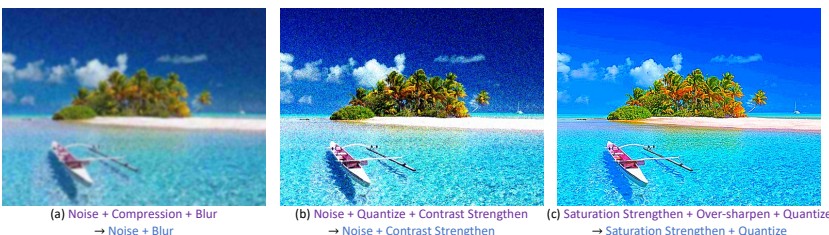

(a) Noise + Compression + Blur → Noise + Blur   (b) Noise + Quantize + Contrast Strengthen → Noise + Contrast Strengthen   (c) Saturation Strengthen + Over-sharpen + Quantize → Saturation Strengthen + Quantize

Figure A3: Humans usually identify at most two distortions (blue) when three (purple) are applied.

## B.2 DETAILS OF DATASET CONSTRUCTION

**Template pool**. As stated in Sec. 3.3, for brief tasks, the questions and answers are templated and sampled from a pool. The questions of detailed tasks are also sampled from a pool. The question pools and answer pools (if possible) of *distortion identification*, *instant rating*, *assessment reasoning*, and *comparison reasoning* tasks are given in Tab. A14, Tab. A16, Tab. A15, and Tab. A17, respectively.

**Statistics of the response length** in our EDQA-495K dataset are detailed in Appendix B.2. We provide statistics on both word count and string length. For the *instant rating* task, there is no distinction between single-distortion and multi-distortion cases. We also depict the word length distribution of detailed reasoning responses in Fig. A4.

**Wordcloud** map of our EDQA-495K dataset is given in Fig. A5. We manually exclude "Image A" and "Image B", since they are constant proper nouns across all texts. The most frequent words in our EDQA-495K dataset (*e.g.*, "overly high", "color quantization", "high contrast", "high saturation", and "detail") are all highly relevant to the low-level properties and the visual quality of images.

## C DETAILS OF MODEL SETUPS

**Model Architecture**. EDQA primarily adopts the architecture from LLaVA-1.6 (Liu et al., 2024) and mPLUG-Owl2 (Ye et al., 2023b), structured as follows. Specifically, the input images and the question texts are first tokenized, then fused, finally processed by the Large Language Model (LLM) for response generation. (1) Tokenizing input images and question texts. We use a frozen CLIP pre-trained ViT-L/14 (Radford et al., 2021) as the image encoder to convert the input images into visual tokens. The question texts are tokenized into textual tokens using the SentencePiece tokenizer (Kudo and Richardson, 2018). To bridge the different embedding spaces of visual and textual tokens, we implement a trainable image abstractor, which is a four-layer transformer network, to map visual tokens into the textual space following Ye et al. (2023b). The abstractor can also significantly reduce the number of vision tokens, relieving the computing pressure. (2) Token fusion. We integrate the visual tokens into pre-defined positions within the textual tokens as token fusion. (3) Response generation using LLM. The fused tokens are fed into LLM to generate the final response. Here we mainly conduct experiments with Vicuna-v1.5-7B. Despite their capabilities, pre-trained

Table A2: **Setting of out-of-distribution (OOD) distortion identification**.

| Category | Training distortions | Validation distortions |
|---|---|---|
| Blur | Motion blur, Glass blur, Lens blur, Zoom blur | Gaussian blur, Jitter blur |
| Noise | Gaussian noise in YCrCb space, Speckle noise, Spatial correlated noise, Poisson noise | Gaussian noise in RGB space, Impulse noise |
| Compression | JPEG compression | JPEG2000 compression |
| Brighten | Shift brighten in HSV & RGB spaces, Gamma brighten in HSV space | Gamma brighten in RGB space |
| Darken | Shift darken in HSV & RGB spaces, Gamma darken in HSV space | Gamma darken in RGB space |
| Contrast strengthen | Contrast strengthen by scaling | Contrast strengthen by stretching |
| Contrast weaken | Contrast weaken by scaling | Contrast weaken by stretching |
| Saturate strengthen | Saturate strengthen in HSV space | Saturate strengthen in YCrCb space |
| Saturate weaken | Saturate weaken in HSV space | Saturate weaken in YCrCb space |
| Quantization | Quantization by OTSU method, Quantization by histogram median | Quantization by histogram equalization |

Table A3: **Response length statistics** in EDQA-495K dataset, reported as word count / string length. For *instant rating* task, there is no distinction between single-distortion and multi-distortion cases.

| | Distortion Identification | Assessment Reasoning | Instant Rating | Comparison Reasoning |
|---|---|---|---|---|
| Single-distortion | 10.36 / 69.81 | 64.37 / 430.23 | 9.30 / 52.02 | 93.20 / 604.97 |
| Multi-distortion | 12.84 / 88.67 | 87.31 / 588.44 | | 114.04 / 740.68 |

LLMs typically do not perform well on IQA tasks without adjustments. Therefore, we employ LoRA (Hu et al., 2021), a fine-tuning technique that efficiently modifies a small subset of parameters within the LLM. Specifically, we apply LoRA to adjust the projection layers in all self-attention modules, following Hu et al. (2021); Yin et al. (2023). This approach allows for targeted refinement of the model's performance on IQA tasks without the need for extensive retraining.

**Model setup**. Since the CLIP pre-trained ViT-L/14 (Radford et al., 2021) encodes each $14 \times 14$ patch to a visual token, the resolution of the input image should be integer multiples of 14. Therefore, we first pad the size of input images to integer multiples of 14 with zero-padding. We encode the image patches into visual tokens using the CLIP pre-trained ViT-L/14 (Radford et al., 2021), with each token having a channel of 1024. The vision abstractor can reduce the number of vision tokens to 64 and map the vision tokens to the hidden dimension of the LLM, which is 4096. Without the vision abstractor, the maximum resolution is limited to 672, constrained by computation resources (RTX A6000 GPUs). However, with the vision abstractor, we can process images with much larger resolutions (up to $2500 \times 2500$). In our experiments, the maximum image resolution is $1092 \times 1456$, thus the resolutions of all images are retained. The vision abstractor consists of four transformer layers with 64 learnable query embeddings. In LoRA of LLM, the parameters of rank and scale factor are both set as 16. There are 32 attention layers in the LLM in total. In each attention layer, the projection weights of "query", "key", "value", and "output" are adjusted using two delta parameters with the shape of $4096 \times 16$ and $16 \times 4096$, respectively.

**Training and inference setup**. In our experiments, we set the LoRA rank to 16. EDQA is trained for 1 epoch with batch size 64. Adam optimizer with $(\beta_1, \beta_2) = (0.9, 0.95)$, weight decay 0.001, and learning rate 0.0002 is used for training. During inference, the temperature is set to 0, since lots of predicted information (*e.g.*, distortion, comparison result) need to be certain.

### C.1 COMPLEXITY AND EFFICIENCY

**Training cost**. The total parameters are 7.11B, including 6.76B for LLM, 0.30B for vision encoder, and 54M for vision abstractor. The trainable parameters are 70M (54M for vision abstractor and 16M for LoRA), constituting only 0.98% of the total parameters. The model is trained on 8 GPUs (RTX A6000). The training is completed in around 22 hours.

**Inference cost**. The inference latency depends on the response length and it is tested on a single RTX A6000 GPU. For example, for brief tasks task with the short answer prompt (about 2.92 words), the inference time stands at approximately 2.23s / batch=32, transformed to 0.07s / sample. For the assessment reasoning task (75.84 words on average), the inference time is 22.97s / batch=32 (*i.e.*, 0.72s per response). EDQA remains deployable on a single consumer GPU (*e.g.*, RTX3090).

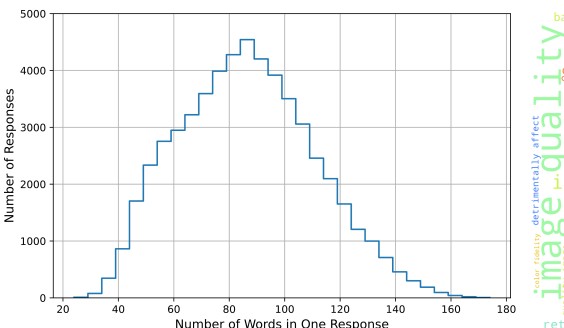

Figure A4: **Word length distribution** of detailed responses in our EDQA-495K dataset.

Figure A5: **Wordcloud** map of our introduced EDQA-495K dataset.

Table A4: **Results of quality score regression** with SRCC / PLCC metrics in full-reference setting.

| Methods | PIPAL$^{test}$ | KADID$^{test}$ | TID2013 | CSIQ |
|---------|----------------|----------------|---------|------|
| SSIM | 0.624 / 0.680 | 0.750 / 0.751 | 0.746 / 0.802 | 0.861 / 0.857 |
| FSIM | 0.673 / 0.746 | 0.855 / 0.857 | 0.841 / 0.875 | **0.937** / 0.937 |
| LPIPS | 0.639 / 0.718 | 0.799 / 0.803 | 0.798 / 0.851 | 0.905 / 0.926 |
| EDQA (Ours) | **0.743 / 0.780** | **0.938 / 0.943** | **0.852 / 0.886** | 0.934 / **0.949** |

# D MORE RESULTS

## D.1 QUALITY SCORE REGRESSION

Our key focus in this work is to *generate descriptive language rather than quality scores*. We focus more on linguistic descriptions because language is an effective interaction tool in an LLM-based intelligent agent. With the rapid development of LLMs and multi-modal techniques, in an LLM-based intelligent agent, language could be a useful tool for interacting and communicating across quality-related tasks such as image assessment, refinement, editing, and recommendation. Still, if it is required, our approach can produce quality scores.

**Quality score regression**. The score regression results are evaluated on the PIPAL, KADID, TID2013, and CSIQ datasets. These datasets include high-quality reference images and their distorted versions under various distortions. We calculate the win rate of an image against others to determine its quality score. Specifically, for an image A, we randomly sample comparison candidates, such as B, C, D, *etc.*, which share the same content as A but have different distortions. Image A is then compared pairwise with each of its comparison candidates (B, C, D, *etc.*). In the full-reference setting, the reference image, Image A, and one compared candidate are input into our model for comparison. In the no-reference setting, only Image A and its one compared candidate are input for comparison. Finally, the win rate of Image A against all its compared candidates is calculated as its quality score. The comparison numbers per image for PIPAL, KADID, TID2013, and CSIQ datasets are 58, 62, 60, and 15, respectively. We show that the comparison number per image could be reduced significantly without large performance degradation in Tab. A7. The results of quality score regression are given in Tab. A4 and Tab. A5, proving that our method can generate accurate quality scores.

**Assessing in-the-wild images with different contents**. Existing real-world IQA datasets like KonIQ Hosu et al. (2020) and SPAQ (Fang et al., 2020) contain real distorted images with various contents. To regress quality scores from such a dataset, our model needs to compare images with different contents though it is trained only to compare images with similar contents, as shown in Task 2 of Fig. 2. The results in Tab. A6 show that even with a task gap between training and test, our original EDQA still achieves comparable results with previous score-based IQA methods in score regression. Furthermore, we formulate real-world IQA datasets into instant rating tasks to re-train our EDQA, *i.e.*, trained on KonIQ then evaluated on SPAQ, and vice versa. Our re-trained EDQA outperforms all baseline score-based methods trained on the same dataset. These results indicate that our method is capable of assessing in-the-wild images with different contents.

Table A5: **Results of quality score regression** with SRCC / PLCC metrics in non-reference setting.

| Methods | PIPAL$^{test}$ | KADID$^{test}$ | TID2013 | CSIQ |
|---|---|---|---|---|
| NIQE | 0.300 / 0.367 | 0.430 / 0.499 | 0.315 / 0.413 | 0.660 / 0.747 |
| CLIPIQA | 0.448 / 0.491 | 0.644 / 0.653 | 0.616 / 0.690 | 0.761 / 0.798 |
| MUSIQ | 0.539 / 0.570 | 0.650 / 0.668 | 0.578 / 0.693 | 0.755 / 0.811 |
| MANIQA | 0.558 / 0.602 | 0.482 / 0.527 | 0.472 / 0.603 | 0.701 / 0.714 |
| EDQA (Ours) | **0.742 / 0.778** | **0.937 / 0.941** | **0.847 / 0.866** | **0.912 / 0.938** |

Table A6: **Results of quality score calculation on SPAQ and KonIQ datasets** with SRCC / PLCC metrics. "LoRA" means the LLM is tuned using LoRA technique, where only 0.24% parameters (16M) of whole LLM (6.76B) are trainable. EDQA needs to *compare images with different contents* to obtain the quality score, since all images in the two datasets contain different contents. The original EDQA is only trained to compare images with similar contents, which brings a task gap. When trained on the same dataset as baselines, EDQA surpasses the baseline methods.

(a) Results on SPAQ dataset

| Methods | NIQE | CLIPIQA | MUSIQ | MANIQA | Q-Align-LoRA | EDQA-LoRA | EDQA-LoRA |
|---|---|---|---|---|---|---|---|
| Train Set | - | - | KonIQ | KonIQ | KonIQ | KonIQ | Original |
| SRCC | 0.664 | 0.700 | 0.856 | 0.755 | 0.854 | **0.859** | 0.835 |
| PLCC | 0.679 | 0.722 | 0.859 | 0.765 | 0.855 | **0.861** | 0.841 |

(b) Results on KonIQ dataset

| Methods | NIQE | CLIPIQA | DBCNN | MUSIQ | Q-Align-LoRA | EDQA-LoRA | EDQA-LoRA |
|---|---|---|---|---|---|---|---|
| Train Set | - | - | SPAQ | SPAQ | SPAQ | SPAQ | Original |
| SRCC | 0.530 | 0.685 | 0.731 | 0.753 | 0.782 | **0.787** | 0.717 |
| PLCC | 0.533 | 0.717 | 0.758 | 0.680 | 0.802 | **0.807** | 0.729 |

**Influence of comparison numbers**. We calculate the win rate of one image over other compared images as the quality score. Here the compared images are selected by round robin for a small number, and random sampling for a large number. For the SPAQ dataset, the number of possible compared images is quite large, thus we adopt the random sampling strategy. The influence of comparison numbers is investigated in Tab. A7. It is shown that the comparison number could be reduced significantly without large performance degradation. In the most extreme cases (*i.e.*, the comparison number is 1 or 2), we use the estimated confidence as weights to calculate the win rate as quality score. Otherwise, the values of the win rate are too discrete (*i.e.*, the values of the win rate can only be 0 or 1 when the comparison number is 1). The results of our EDQA are still reasonable in such extreme cases. Considering that the random sampling may bring large randomness or variance, we average the results with 5 random runs for small comparison numbers (*i.e.*, $< 10$). Although the comparison number is small and the sampling process is random, our method is still very stable with relatively small standard deviations in Tab. A7.

## D.2 MORE ABLATION STUDIES

**Ablation study on retaining resolution**. In Tab. 9 of the main paper, the effects of retaining resolution is studied. Specially, we randomly sample 1,000 high-quality images whose aspect ratios are greater than $4:3$. These images are either resized by swapping their height and width (denoted as H↔W), or bi-linearly down-sampled by a scale factor of 0.5, 0.75, 0.8, 0.85, 0.9, or 0.95. EDQA is requested to conduct the *instant rating* task, *i.e.*, compare the original and resized images to determine the superior one. Note that in Tab. A8, the alternative method of retaining resolution is to resize both original image and resized image to a larger resolution, which can maintain the quality difference. In contrast, resizing both images to smaller resolution results in two nearly same images.

The results are presented in Tab. A8. First, overall, retaining resolution is crucial for identifying images with better aspect ratio or higher resolution. Second, with down-sampling becomes severer (*i.e.*, aspect ratio is from 0.95 to 0.5), the accuracy is improved since the quality drop is more significant. Third, for severe down-sampling (*e.g.*, aspect ratio is 0.5) where the quality degradation is quite obviously, retaining resolution or just resizing both images to a larger size both perform well

Table A7: **Influence of comparison numbers per image on SPAQ dataset** with SRCC and PLCC metrics. "(KONIQ)" means the model is trained on KONIQ dataset, which is also an in-the-wild IQA dataset. For small comparison numbers ($< 10$), we average the results with 5 random runs.

| Comparison Numbers | | 100 | 50 | 25 | 10 | 5 | 2 | 1 |
|---|---|---|---|---|---|---|---|---|
| EDQA (Original) | SRCC | 0.835 | 0.832 | 0.826 | 0.806 | 0.731±0.006 | 0.647±0.009 | 0.577±0.015 |
| | PLCC | 0.841 | 0.837 | 0.832 | 0.810 | 0.735±0.006 | 0.639±0.009 | 0.537±0.015 |
| EDQA (KONIQ) | SRCC | 0.859 | 0.854 | 0.850 | 0.830 | 0.756±0.006 | 0.664±0.011 | 0.598±0.013 |
| | PLCC | 0.861 | 0.858 | 0.852 | 0.833 | 0.757±0.007 | 0.652±0.011 | 0.546±0.015 |

Table A8: **Retaining resolution** during both training and inference is important to identify images with better aspect ratio or higher resolution.

| Retain Resolution? | | H↔W | 0.5× | 0.75× | 0.8× | 0.85× | 0.9× | 0.95× |
|---|---|---|---|---|---|---|---|---|
| Training | Inference | | | | | | | |
| ✗ | ✗ | 73.0 | 99.0 | 93.5 | 91.7 | 83.8 | 77.2 | 71.2 |
| ✓ | ✗ | 85.6 | 99.8 | 99.4 | 99.0 | 95.9 | 94.8 | 89.4 |
| ✓ | ✓ | 98.8 | 99.9 | 99.6 | 99.3 | 99.1 | 96.8 | 97.0 |

($\leq 99.0$). Finally, however, for relatively slight down-sampling (*e.g.*, aspect ratio is from 0.75 to 0.95), the performance of retaining resolution is stably superior than resizing.

**Ablation study on confidence estimation**. We further examine the correlation between model performance and estimated confidence scores on a wider range of benchmarks. The results are illustrated in Fig. A6. The performance of our model is consistently enhanced as the confidence interval increases, validating the effectiveness of our confidence estimation.

**Ablation study on model architecture**. We compare two vision-text connectors (*i.e.*, vision abstractor *v.s.* projector) and three LLMs on distortion identification and instant rating tasks. The default vision-text connector and LLM in this ablation study is vision projector and Vicuna-v1.5-7b. The results in Tab. A9 show that the performance is similar. Considering that vision abstractor can greatly reduce the computational burden than projector, we select abstractor in the main paper. For example, for a $448 \times 448$ image, projector generates 1024 tokens, while abstractor only outputs 64 tokens. Note that the amount of computation is proportional to the square of the number of tokens.

**Relationships between the comparison reasoning and instant rating tasks** are studied in Tab. A10. First, comparison reasoning task improves the performance on four instant rating datasets, but decreases the results on two datasets. Overall, comparison reasoning task helps the instant rating. Second, instant rating task stably improves the performance on comparison reasoning task.

**Influence of input order on quality comparison**. Recently, Zhu et al. (2024b) shows that some VLMs are sensitive to the input order of paired images when comparing the quality of two images. We test our model on the fine-grained dataset released by Zhu et al. (2024b) in Tab. A11. We follow Zhu et al. (2024b) to report the consistency / accuracy / correlation as metrics. Consistency means consistency in changing the order of input images. Our model achieves more than 0.90 consistency in all splits. Also, our comparison accuracy and score correlation are both much higher than Q-Instruct and GPT-4V. These results show that our model is robust to the order of input images. The statistics of our model's confidence are given in Tab. A12. The results show that the confidence of consistent prediction is much higher than inconsistent prediction, reflecting the self-evaluation ability of EDQA.

### D.3 DETAILS OF REAL-WORLD APPLICATIONS

**Details of quality comparison on model-processed images**. We consider five common image restoration tasks: super-resolution, denoising, JPEG compression artifact removal, motion deblurring, and defocus deblurring. For each task, we collect three to four cutting-edged models in recent years (listed in Tab. A13), apply them to a correspondingly degraded image, and then manually rank the resultant model-processed images. To find the image

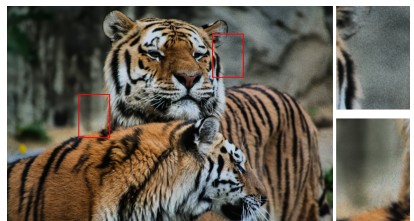

Figure A7: An example of the model-restored image.

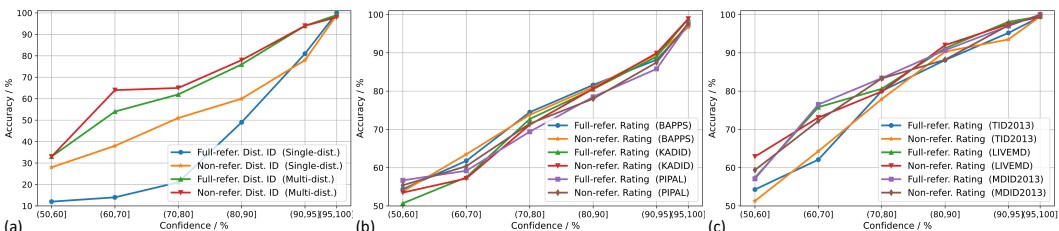

Figure A6: **Our estimated confidence scores** are high correlated to the model performance on (a) *distortion identification* and (b) (c) *instant rating* tasks on different benchmarks in both full-reference and non-reference settings.

Table A9: **Model architecture** (*i.e.*, vision-text connectors and LLMs) has relatively little influence on model performance.

| Types | Architectures | Distortion Identification | | Instant Rating | | | | | |
|---|---|---|---|---|---|---|---|---|---|
| | | Single-dist. ID | Multi-dist. ID | BAPPS$^{test}$ | KADID$^{test}$ | PIPAL$^{test}$ | TID2013 | LIVE-MD | MDID2013 |
| Vision-text connector | Projector | 97.9 / 94.7 | 90.5 / 89.5 | 82.7 / 81.4 | 92.7 / 92.4 | 89.2 / 88.8 | 96.2 / 95.9 | 92.1 / 91.9 | 89.1 / 88.4 |
| | Abstractor | 97.7 / 94.1 | 91.3 / 89.3 | 84.7 / 82.4 | 93.6 / 93.1 | 90.5 / 90.0 | 96.9 / 96.4 | 92.1 / 91.8 | 90.0 / 89.6 |
| LLM | Vicuna-v1.5-7b | 97.9 / 94.7 | 90.5 / 89.5 | 82.7 / 81.4 | 92.7 / 92.4 | 89.2 / 88.8 | 96.2 / 95.9 | 92.1 / 91.9 | 89.1 / 88.4 |
| | Vicuna-v0-7b | 96.9 / 93.6 | 89.8 / 89.3 | 82.5 / 81.3 | 92.8 / 92.2 | 88.2 / 88.1 | 95.0 / 94.7 | 91.2 / 91.1 | 90.5 / 90.2 |
| | LLaMA-2-7b | 97.0 / 94.0 | 90.6 / 89.1 | 81.7 / 81.5 | 92.6 / 92.0 | 88.4 / 87.9 | 94.6 / 94.1 | 91.8 / 91.1 | 90.9 / 90.7 |

considered best by VLMs, we linearly scan the candidates and compare them in pairs. As VLMs' results are not deterministic and may be sensitive to the presentation order of images, we repeat the linear scan 10 times and randomly shuffle the scan order each time.

We have shown that our EDQA can consistently find the near-optimal candidate compared to GPT-4V and scored-based methods. A highlight is that our EDQA generalizes well on these out-of-distribution (OOD) model-processed images. For example, the image in Fig. A7 is restored from a noisy image. There is still remnant noise, which is somewhat strange. For such an OOD image, our EDQA correctly recognizes it to be inferior, but MANIQA, MUSIQ, and NIQE consider it as the best of the four candidates. We provide two qualitative results of detailed comparison reasoning on model-processed images in Fig. A8. One compares SwinIR and FBCNN in the JPEG compression artifact removal task, and the other compares MPRNet and MAXIM in the deblur task. Our model can generate a reasonable explanation for the comparison results.

## D.4    MORE QUALITATIVE RESULTS

More qualitative results of *assessment reasoning*, *comparison reasoning*, and assessment on web-downloaded images are presented in Fig. A10, Fig. A11, Fig. A12, Fig. A13, and Fig. A14. EDQA could accurately identify distortions, analyze their impacts on the display of image contents, then weigh the advantages and disadvantages of different aspects, and finally draw a final conclusion (*e.g.*, overall quality, comparison results). In Fig. A9, we also present some qualitative results on assessing images with extremely severe distortions. Our model performs well in such extreme cases.

Table A10: **Relationships between the comparison reasoning task and instant rating task**. Overall, these two tasks are beneficial to each other.

(a) Results on the instant rating task

|  | BAPPS$^{test}$ | KADID$^{test}$ | PIPAL$^{test}$ | TID2013 | LIVE-MD | MDID2013 | Mean |
|---|---|---|---|---|---|---|---|
| Only Rating | 81.6 / 81.6 | 92.4 / 92.3 | 89.1 / 89.0 | 94.2 / 94.1 | **92.9 / 92.7** | **92.1 / 91.7** | 90.4 / 90.2 |
| Co-training | **84.7 / 82.4** | **93.6 / 93.1** | **90.5 / 90.0** | **96.9 / 96.4** | 92.1 / 91.8 | 90.0 / 89.6 | **91.3 / 90.6** |

(b) Results on the comparison reasoning task

|  | Single-distortion | | | Multi-distortion | | |
|---|---|---|---|---|---|---|
|  | GPT-4 Score | BLEU | ROUGE-L | GPT-4 Score | BLEU | ROUGE-L |
| Only Reasoning | 74.3 / 69.6 | 0.203 / 0.202 | 0.465 / 0.453 | 70.6 / **69.1** | 0.165 / 0.165 | 0.414 / 0.407 |
| Co-training | **75.1 / 74.9** | **0.207 / 0.207** | **0.466 / 0.463** | **71.7** / 68.7 | **0.176 / 0.172** | **0.420 / 0.413** |

Table A11: **Quality comparison results** on the fine-grained dataset released by Zhu et al. (2024b) with consistency / accuracy / correlation metrics.

| Datasets | Setting | Q-Instrcut | GPT-4V | EDQA (Ours) |
|---|---|---|---|---|
| CSIQ | Various levels, same type | 0.115 / 0.081 / 0.557 | 0.419 / 0.402 / 0.906 | **0.955 / 0.925 / 0.958** |
| CSIQ | Various types, same level | 0.117 / 0.069 / 0.416 | 0.325 / 0.244 / 0.482 | **0.905 / 0.690 / 0.857** |
| SPAQ | Various score regions | 0.448 / 0.233 / 0.328 | 0.653 / 0.398 / 0.448 | **0.921 / 0.596 / 0.961** |

Table A12: **Confidence statistics** on the fine-grained dataset released by Zhu et al. (2024b) within consistent and inconsistent responses.

| Datasets | CSIQ (various levels) | | CSIQ (various types) | | SPAQ | |
|---|---|---|---|---|---|---|
|  | Consistent | Inconsistent | Consistent | Inconsistent | Consistent | Inconsistent |
| Confidence | 0.933±0.120 | 0.629±0.083 | 0.860±0.141 | 0.611±0.098 | 0.900±0.125 | 0.649±0.110 |

Table A13: **Image restoration models** used in comparing model-processed images, including SwinIR (Liang et al., 2021), HAT (Chen et al., 2023b), X-Restormer (Chen et al., 2023a), MPRNet (Zamir et al., 2021), Restormer (Zamir et al., 2022), FBCNN (Jiang et al., 2021), MAXIM (Tu et al., 2022), MPRNet (Zamir et al., 2021), DRBNet (Ruan et al., 2022), and IFAN (Lee et al., 2021). For FBCNN, "$q$=90" means training on the quality factor 90, and "blind" means blind to the quality factor.

| Image restoration task | Image restoration models |
|---|---|
| Super-resolution | SwinIR, HAT, X-Restormer |
| Denoising | SwinIR, MPRNet, Restormer, X-Restormer |
| JPEG compression artifact removal | SwinIR, FBCNN ($q$=90), FBCNN (blind) |
| Motion deblurring | MAXIM, MPRNet, Restormer |
| Defocus deblurring | DRBNet, IFAN, Restormer |

JPEG compression artifact removal

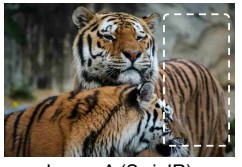 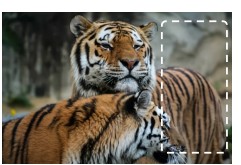

Image A (SwinIR)     Image B (FBCNN)

> The images depict two tigers, one with orange and black stripes and the other with a mix of orange and white stripes, with a natural background.
>
> In Image A, there is a slight compression artifact present, which can be observed as a subtle blockiness in the image, reducing the sharpness and detail, particularly noticeable in the fur texture of the tigers. Image B appears to be free from such distortions, maintaining the integrity of the details and colors.
>
> Comparing the quality of Image A and Image B, Image B is of better quality. It retains the clarity and detail of a high-quality image, while Image A's slight compression slightly degrades the visual information.

Deblur

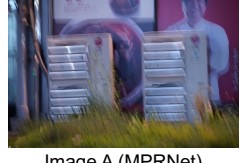 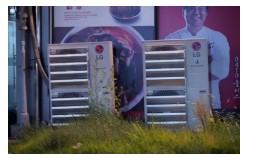

Image A (MPRNet)     Image B (MAXIM)

> The images depict an outdoor scene with air conditioning units in the foreground and a large poster in the background.
>
> In Image A, there is a slight blur distortion, reducing the sharpness and detail of the image content, particularly noticeable on the text and finer details of the poster and units. Image B appears to be free from distortion, maintaining clarity and detail throughout.
>
> Comparing the two, Image B has better quality as it retains the sharpness and detail of a high-quality image, making it superior to Image A, which suffers from a slight loss of detail due to the blur distortion.

Figure A8: **Qualitative results** of detailed comparison reasoning on model-processed images.

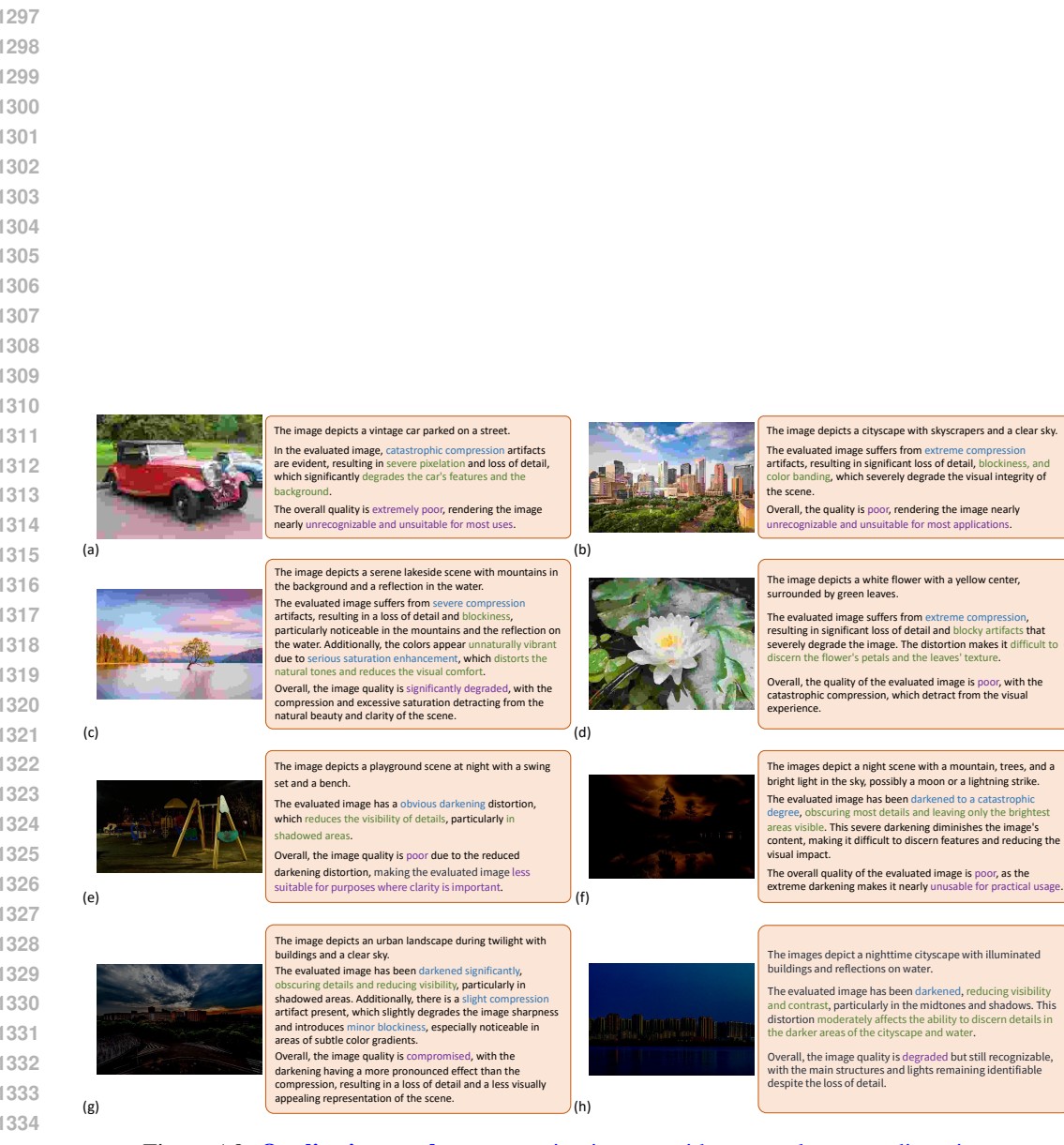

Figure A9: **Qualitative results** on assessing images with extremely severe distortions.

Table A14: Question pool and answer pool of *distortion identification* task.

| # | Question / Answer |
|---|---|
| 1 | Q: What are the primary degradation(s) observed in the evaluated image?
A: The primary degradation(s) in the evaluated image is/are {}. |
| 2 | Q: What distortion(s) are most apparent in the evaluated image?
A: The most apparent distortion(s) in the evaluated image is/are {} |
| 3 | Q: Identify the chief degradation(s) in the evaluated image.
A: The chief degradation(s) in the evaluated image is/are {}. |
| 4 | Q: Pinpoint the foremost image quality issue(s) in the evaluated image.
A: The foremost image quality issue(s) is/are {}. |
| 5 | Q: What distortion(s) stand out in the evaluated image?
A: The distortion(s) that stand out is/are {}. |
| 6 | Q: What distortion(s) are most prominent in the evaluated image?
A: The most prominent distortion(s) is/are {}. |
| 7 | Q: What critical quality degradation(s) are present in the evaluated image?
A: The critical quality degradation(s) presented is/are {}. |
| 8 | Q: Highlight the most significant distortion(s) in the evaluated image.
A: The most significant distortion(s) in the evaluated image is/are {}. |
| 9 | Q: What distortion(s) most detrimentally affect the overall quality of the evaluated image?
A: The distortion(s) that most detrimentally affect the overall quality is/are {}. |
| 10 | Q: Determine the most impactful distortion(s) in the evaluated image.
A: The most impactful distortion(s) in the evaluated image is/are {}. |
| 11 | Q: Identify the most notable distortion(s) in the evaluated image's quality.
A: The most notable distortion(s) in the evaluated image's quality is/are {}. |
| 12 | Q: What distortion(s) most significantly affect the evaluated image?
A: The distortion(s) that most significantly affect the evaluated image is/are {}. |
| 13 | Q: Determine the leading degradation(s) in the evaluated image.
A: The leading degradation(s) is/are {}. |
| 14 | Q: What distortion(s) are most prominent when examining the evaluated image?
A: The most prominent distortion(s) is/are {}. |
| 15 | Q: What distortion(s) are most evident in the evaluated image?
A: The most evident distortion(s) in the evaluated image is/are {}. |
| 16 | Q: What quality degradation(s) are most apparent in the evaluated image?
A: The most apparent quality degradation(s) is/are {}. |
| 17 | Q: In terms of image quality, what are the most glaring issue(s) with the evaluated image?
A: The most glaring issue(s) with the evaluated image is/are {}. |
| 18 | Q: What are the foremost distortion(s) affecting the evaluated image's quality?
A: The foremost distortion(s) affecting the evaluated image's quality is/are {}. |
| 19 | Q: Identify the most critical distortion(s) in the evaluated image.
A: The most critical distortion(s) is/are {}. |
| 20 | Q: In the evaluated image, what distortion(s) are most detrimental to image quality?
A: In the evaluated image, {} is/are the most detrimental distortion(s) to image quality. |
| 21 | Q: What are the most severe degradation(s) observed in the evaluated image?
A: The most severe degradation(s) is/are {}. |
| 22 | Q: What are the leading distortion(s) in the evaluated image?
A: The leading distortion(s) in the evaluated image is/are {}. |
| 23 | Q: What are the most critical image quality issue(s) in the evaluated image?
A: The most critical image quality issue(s) in the evaluated image is/are {}. |
| 24 | Q: What distortion(s) most notably affect the clarity of the evaluated image?
A: The distortion(s) that most notably affect the clarity is/are {}. |

Table A15: Question pool of *assessment reasoning* task.

| # | Question |
|---|---|
| 1 | Could you assess the overall quality of the image and elaborate on your evaluation? |
| 2 | How would you rate the image's quality, and what factors contribute to your assessment? |
| 3 | Can you provide a detailed evaluation of the image's quality? |
| 4 | Please evaluate the image's quality and provide your reasons. |
| 5 | How do you perceive the quality of the image, and what aspects influence your judgment? |
| 6 | Offer an assessment of the image's quality, highlighting any strengths or weaknesses. |
| 7 | What is your opinion on the quality of the image? Explain your viewpoint. |
| 8 | Assess the quality of the image with detailed reasons. |
| 9 | How does the image's quality impact its overall effectiveness or appeal? |
| 10 | Evaluate the image's quality and justify your evaluation. |
| 11 | How about the overall quality of the image, and why? |
| 12 | Provide a thorough evaluation of the image's quality. |
| 13 | Examine the image's quality by considering factors influencing its clarity. |
| 14 | Analyze the image's quality, and detail your findings. |
| 15 | Provide a comprehensive assessment of the image's quality, including both strengths and areas for improvement. |
| 16 | Assess the image's quality from a professional standpoint. |
| 17 | Evaluate the image's clarity and explain how it contributes to the overall quality. |
| 18 | How would you rate the overall quality of the image, and why? |
| 19 | What is your opinion on the image's quality? Elaborate on your evaluation. |
| 20 | Evaluate the quality of the image and provide a comprehensive explanation. |

Table A16: Question pool and answer pool of *instant rating* task.

| # | Question / Answer |
|---|---|
| 1 | Q: Which image do you believe has better overall quality: Image A or Image B? 
 A: I believe Image {} has better overall quality. |
| 2 | Q: Determine which image exhibits higher quality between Image A and Image B. 
 A: In my assessment, Image {} exhibits higher quality. |
| 3 | Q: Compare the general quality of Image A and Image B, and state your preference. 
 A: My preference leans towards Image {} to have better general quality. |
| 4 | Q: In your opinion, which image demonstrates superior quality: Image A or Image B? 
 A: In my opinion, Image {} demonstrates superior quality. |
| 5 | Q: Which of the two images, Image A or Image B, do you consider to be of better quality? 
 A: I consider Image {} to be of better quality. |
| 6 | Q: Evaluate the quality of Image A and Image B, and decide which one is superior. 
 A: I conclude that Image {} is superior. |
| 7 | Q: Between Image A and Image B, which image do you think has better quality overall? 
 A: I think Image {} has better quality overall. |
| 8 | Q: Determine which image, Image A or Image B, you perceive to have better quality. 
 A: I determine that Image {} has better quality. |
| 9 | Q: Assess the quality of Image A and Image B, and choose the one you believe is superior. 
 A: I choose Image {} to be superior in terms of quality. |
| 10 | Q: Which image stands out to you as having better quality: Image A or Image B? 
 A: Image {} stands out as the superior choice in terms of quality. |
| 11 | Q: Can you compare the quality of Image A and Image B and decide which one is better? 
 A: I find Image {} to be better after comparing the quality of both. |
| 12 | Q: Decide which image, Image A or Image B, you think possesses higher quality. 
 A: I decide that Image {} possesses higher quality. |
| 13 | Q: Evaluate Image A and Image B, and select the one that you feel has better quality. 
 A: Upon evaluation, I select Image {} as the one with better quality. |
| 14 | Q: Which of the two images, Image A or Image B, appears to have superior quality to you? 
 A: To me, Image {} appears to have superior quality. |
| 15 | Q: Compare the quality of Image A and Image B, and determine which one you prefer. 
 A: My preference leans towards Image {} after comparing the quality. |
| 16 | Q: Make a judgment on which image, Image A or Image B, you consider to be of better quality. 
 A: I consider Image {} to be of better quality. |
| 17 | Q: Between Image A and Image B, which image do you perceive to have better quality overall? 
 A: I perceive Image {} to have better quality overall. |
| 18 | Q: Assess the quality of Image A and Image B, and indicate which one you find to be better. 
 A: I find Image {} emerges as the better option with superior quality. |
| 19 | Q: Which image, Image A or Image B, do you think displays better quality when compared? 
 A: When compared, Image {} displays better quality. |
| 20 | Q: Differentiate between Image A and Image B in terms of overall quality and decide which one is superior. 
 A: Image {} differentiates itself with superior quality. |

Table A17: Question pool of *comparison reasoning* task.

| # | Question |
|---|---|
| 1 | Compare the overall quality of Image A with Image B and provide a comprehensive explanation. |
| 2 | Which image has better visual quality, Image A or Image B? Can you explain the comparison results? |
| 3 | Evaluate the general visual appeal and quality of both Image A and Image B, and elaborate on which one excels. |
| 4 | Discuss the overall impression and quality of Image A versus Image B, and justify your assessment. |
| 5 | Compare the overall quality between Image A and Image B, and justify your comparison results. |
| 6 | Assess the overall visual quality of Image A and Image B, discussing which one delivers a more compelling visual quality. |
| 7 | Which image demonstrates higher overall quality, Image A or Image B? Please provide detailed reasoning for your evaluation. |
| 8 | Analyze the overall quality of both Image A and Image B, and explain which image stands out. |
| 9 | Compare the perceived quality of Image A with Image B, providing insights into their respective strengths and weaknesses. |
| 10 | Discuss the visual quality of Image A and Image B, and elaborate on which one appears more appealing. |
| 11 | Can you evaluate the overall quality in both Image A and Image B, and explain which one is superior? |
| 12 | Compare the overall visual impact and impression of Image A versus Image B, and justify your assessment of their quality. |
| 13 | Which image exhibits higher overall quality: Image A or Image B? Please explain your reasoning. |
| 14 | Evaluate the visual quality in Image A and Image B, providing insights into their comparative strengths. |
| 15 | Compare the overall quality between Image A and Image B, and discuss which one appears more appealing. |
| 16 | Assess the visual quality of both Image A and Image B, and explain which one is better. |
| 17 | Which image demonstrates superior quality: Image A or Image B? Please elaborate on your evaluation. |
| 18 | Discuss the overall impression of Image A versus Image B, and justify your assessment of their comparative quality. |
| 19 | Compare the visual quality of Image A with Image B, providing detailed insights into their respective strengths and weaknesses. |
| 20 | Evaluate the overall quality of Image A and Image B, and explain which one has higher quality. |

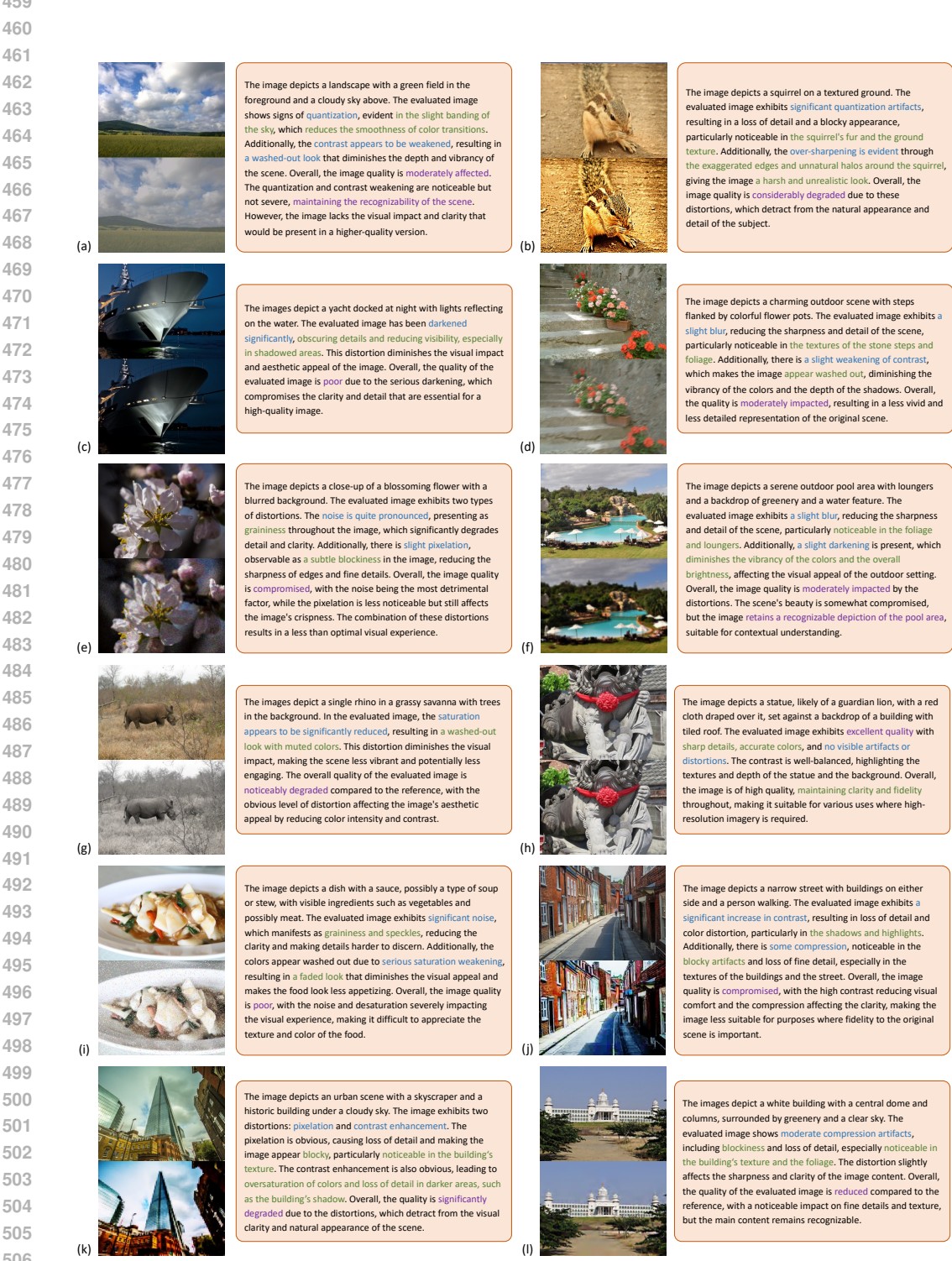

Figure A10: **Qualitative results** on *assessment reasoning* task in the full-reference setting. The two images from top to down are the reference image and evaluated image, respectively.

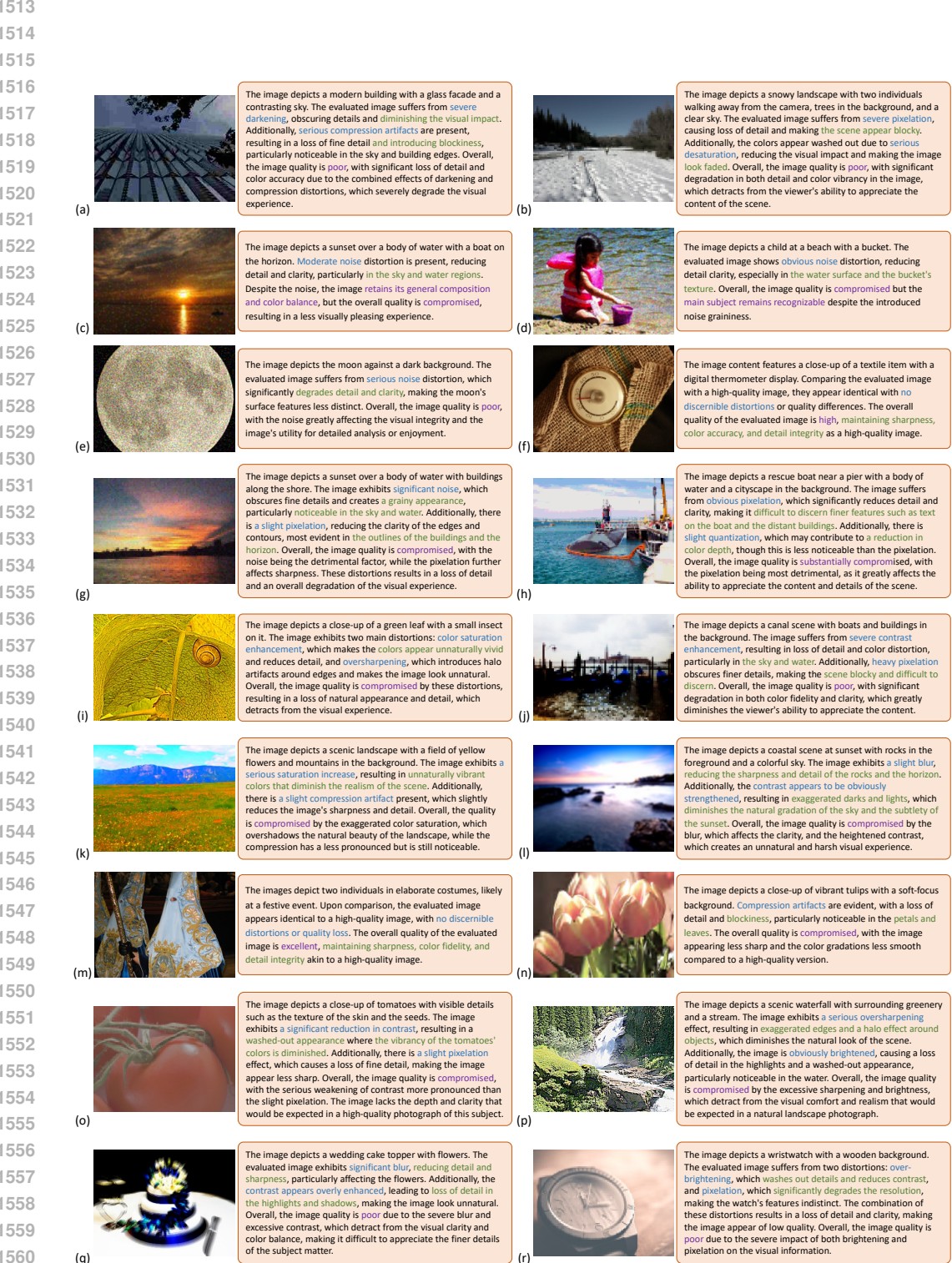

Figure A11: **Qualitative results** on *assessment reasoning* task in the non-reference setting.

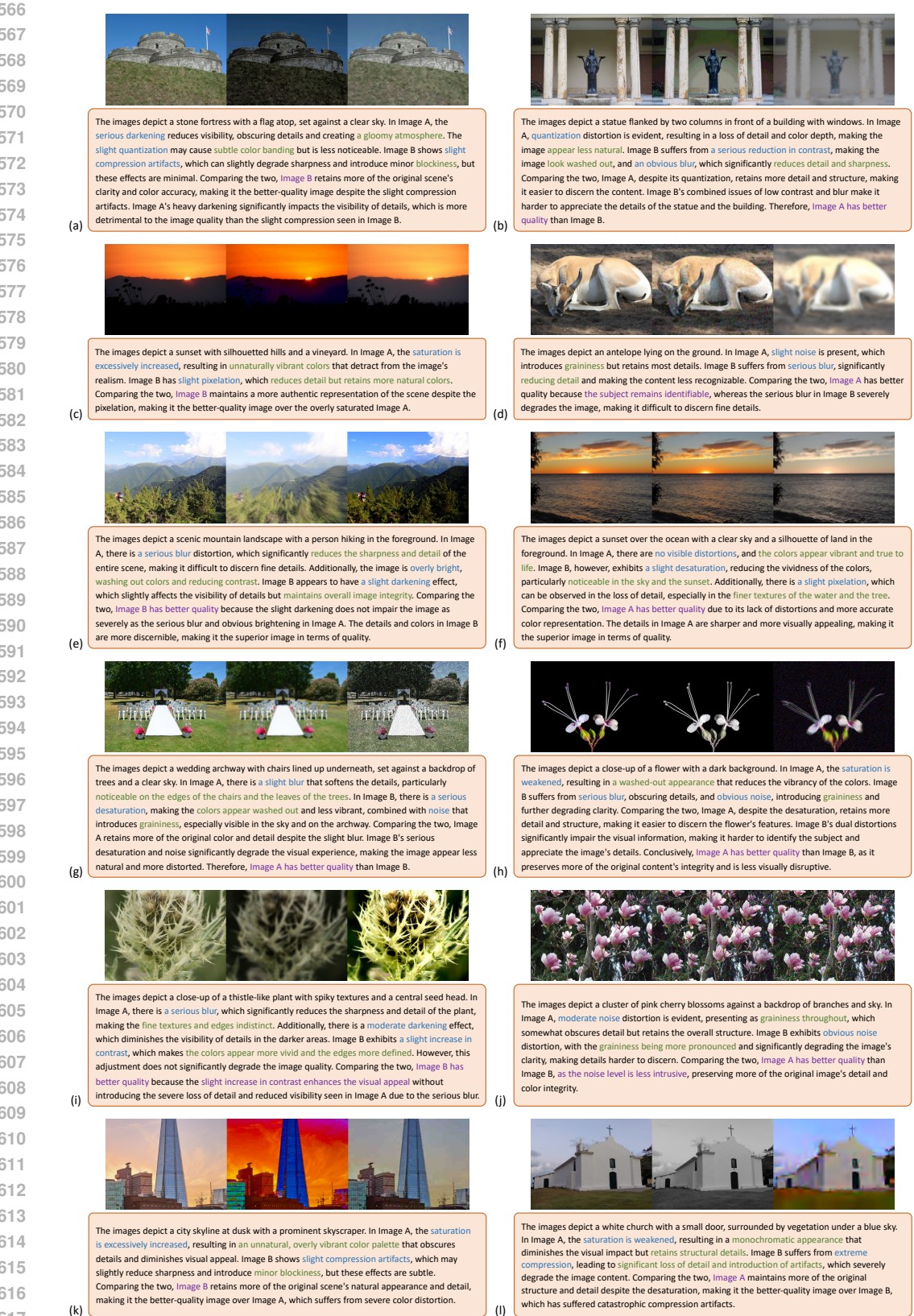

Figure A12: **Qualitative results** on *comparison reasoning* task in the full-reference setting. The three images from left to right are the reference image, Image A, and Image B, respectively.

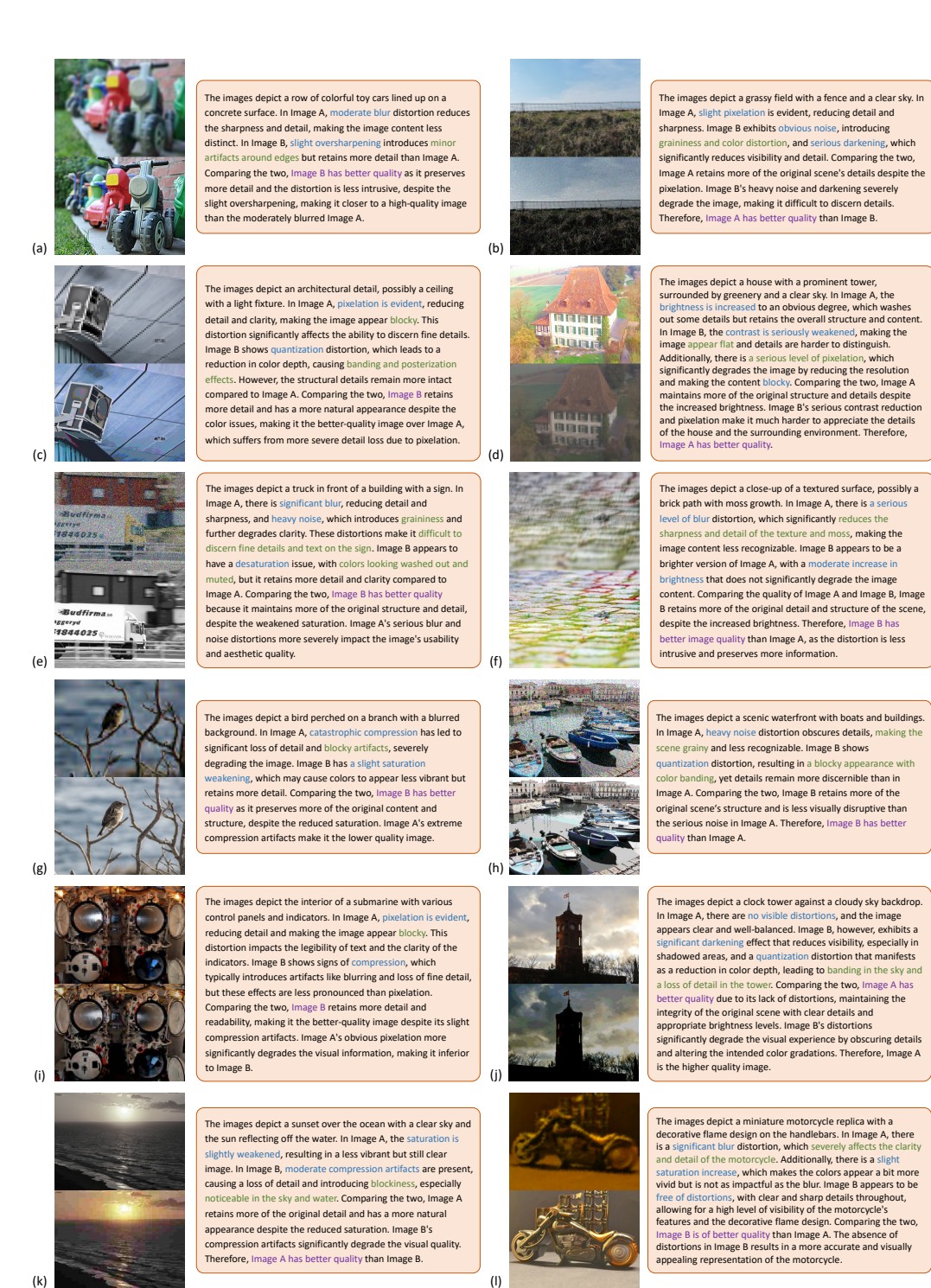

Figure A13: **Qualitative results** on *comparison reasoning* task in the non-reference setting. The two images from top to down are Image A and Image B, respectively.

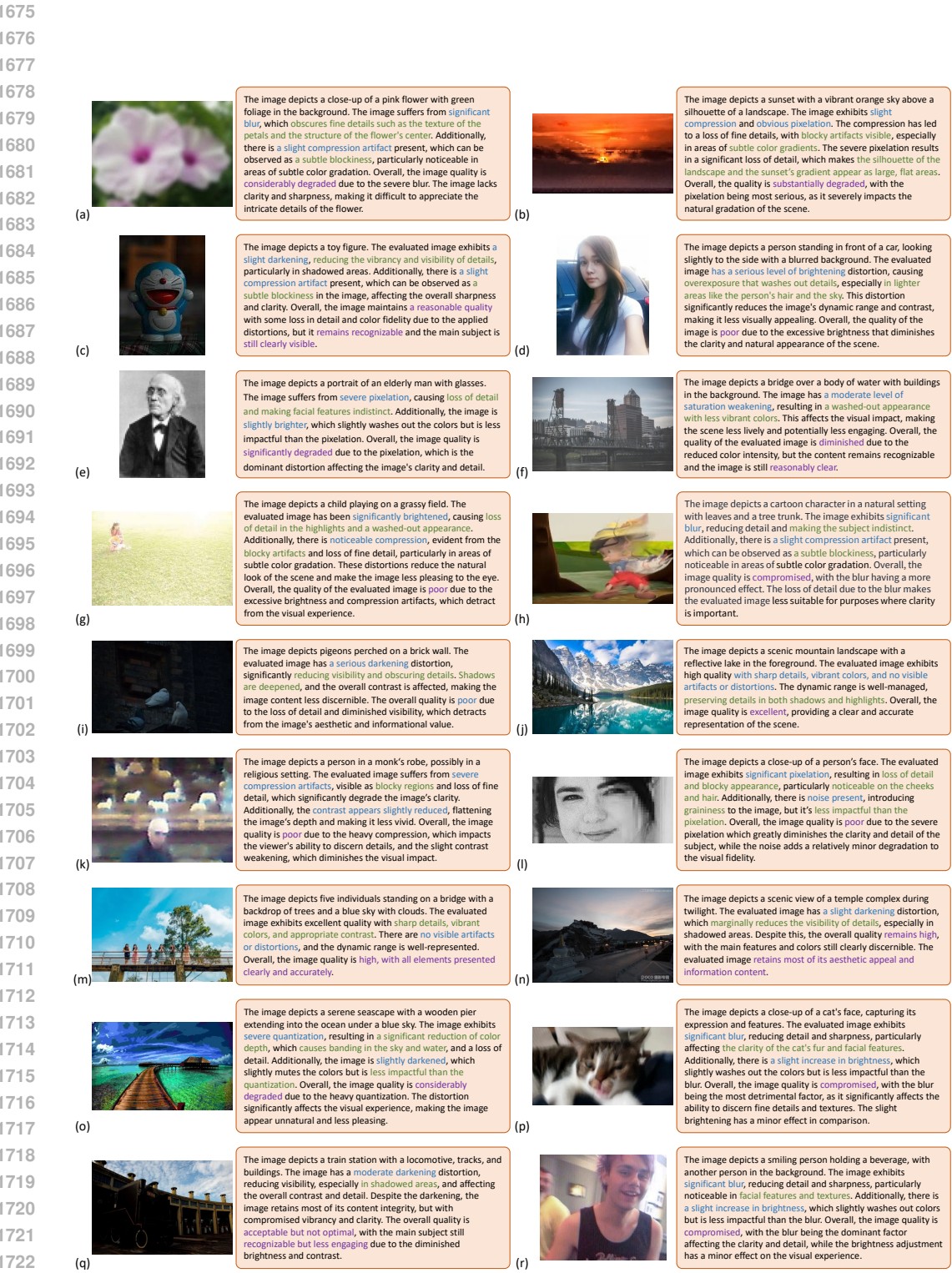

Figure A14: **Qualitative results** on assessing web-downloaded images.

