# OpenReview forum: "Enhancing Descriptive Image Quality Assessment with a Large-scale Multi-modal Dataset"
_ICLR.cc/2025/Conference — Submitted to ICLR 2025_

### Official Review · Reviewer_5EhH · 2024-10-29

**Soundness:** 2
**Presentation:** 3
**Contribution:** 2
**Rating:** 6
**Confidence:** 3

**Summary:**

In this paper, the authors propose Enhanced Descriptive image Quality Assessment (EDQA), a multi-functional IQA paradigm that addresses limitations in current VLM-based IQA methods. EDQA supports diverse tasks, including both full-reference and non-reference IQA scenarios and incorporates EDQA-495K—a large, high-quality dataset informed by ground-truth data. With image resolution retained, EDQA can handle resolution-dependent quality issues effectively, and it offers confidence scores to filter low-quality outputs.

**Strengths:**

1. The proposed model supports both full-reference and non-reference IQA tasks.
2. A large-scale dataset with 495k images and annotations is constructed, which will benefit VLM-based IQA model training.
3. Key tokens are analyzed for confidence score estimation, indicating the model’s uncertainty in its responses.

**Weaknesses:**

1. Only synthetic distortion types are included in the dataset construction. Although the authors test performance on the SPAQ dataset, inferior results compared to existing VLM models (e.g., Q-Align) raise concerns about the model’s functionality. Additional testing on authentic distortions is recommended to verify performance.
2. The paper lacks an investigation of comparison order effects. In full-reference IQA, the order of images (e.g., reference first, then distorted, or vice versa) significantly impacts model performance, and it’s crucial to verify model robustness in this scenario. The authors could consider using a small set of existing datasets or testing on the datasets released in [2AFC Prompting of Large Multimodal Models for Image Quality Assessment, TCSVT 2024] to strengthen performance validation.
3. While the authors state that image resolution is retained, the appendix reveals that images larger than 672 pixels are resized bilinearly, potentially diminishing the claimed resolution advantage. It is recommended that the authors refine this claim for accuracy.
4. Distortion severity in the constructed dataset appears limited. In real scenarios, distortion can be more severe (e.g., extreme low-light conditions), raising concerns about the model's generalization capability to such challenging, real-world conditions. It would be beneficial for the authors to test the model in real-world scenarios, including severe compression, low-light conditions, and adverse weather such as fog and rain, to better evaluate its practical applicability.

**Questions:**

1. An investigation into comparison order effects should be conducted to verify the model’s robustness.
2. The model’s generalization capacity on distorted images in the wild should be examined.

---

> ### Author Response · Authors · 2024-11-21
> **Responses to Reviewer 5EhH**
>
> Thanks to the reviewer's valuable reviews and appreciation of our dataset contribution, comprehensive task paradigm, and analysis of model confidence. We address the concerns point-by-point below.
>
> **Q1. More evaluation of in-the-wild images.**
>
> We test our model on two in-the-wild datasets, SPAQ and KonIQ. The results are shown below and added to Table A6 of the revised version.
>   - First, directly evaluating our original EDQA on KonIQ and SPAQ, we achieve comparable results with previous score-centered IQA methods. Note that our EDQA is only trained to compare images with similar contents, as shown in Task 2 of Fig. 2. However, to calculate the quality scores of these two datasets, our model needs to compare images with different contents, since all images in these two datasets contain different contents.
>   - Second, after finetuning our EDQA on real-world IQA datasets, we outperform baseline methods. We adopt the same training and evaluation setting of baseline methods, i.e., trained on KonIQ then evaluated on SPAQ, and vice versa. To retrain, we sample image pairs from the real-world IQA dataset to formulate instant rating pairs.
>   - Additionally, we finetune Q-Align using the same LoRA parameters as our model to only tune 0.24% LLM parameters. The performance of Q-Align is slightly inferior to EDQA. This shows that the advantage of Q-Align comes from model size, i.e., fully tuning the whole LLM.
>
> | Results on SPAQ | NIQE | CLIPIQA | DBCNN | MUSIQ | ManIQA | Q-Align-LoRA | EDQA-LoRA | EDQA-LoRA |
> |-----|-----|-----|-----|-----|-----|-----|-----|-----|
> | Train Set | -|-| KonIQ | KonIQ | KonIQ | KonIQ | KonIQ | Original |
> | SRCC| 0.664 | 0.700 | 0.806 | 0.856 | 0.755 | 0.854 | **0.859** | 0.835 |
> | PLCC| 0.679 | 0.722 | 0.812 | 0.859 | 0.765 | 0.855 | **0.861** | 0.841 |
>
> | Results on KonIQ | NIQE| CLIPIQA | NIMA | DBCNN   | MUSIQ   | Q-Align-LoRA | EDQA-LoRA| EDQA-LoRA|
> |-----|-----|-----|-----|-----|-----|-----|-----|-----|
> | Train Set| - | - | SPAQ | SPAQ | SPAQ | SPAQ | SPAQ | Original |
> | SRCC | 0.530 | 0.685 | 0.733 | 0.731 | 0.753   | 0.782 | **0.787** | 0.717 |
> | PLCC | 0.533 | 0.717 | 0.788 | 0.758 | 0.680   | 0.802 | **0.807** | 0.729 |
>
> **Q2. Will the performance drop if switching the order of two input images?**
>
> Thanks for the suggestion. The table below shows that our model achieves more than 0.90 consistency if switching the order of two input images. Also, our comparison accuracy and score correlation are both much higher than Q-Instruct and GPT-4V.
>
> We test our model on the fine-grained dataset released by [R1]. We follow [R1] to report the consistency / accuracy / correlation as metrics, where consistency means the consistency in changing the order of input images. The results of Q-Instruct and GPT-4V are borrowed from [R1]. These results show that our model is robust to the order of input images.
>
> [R1] 2AFC Prompting of Large Multimodal Models for Image Quality Assessment, TCSVT 2024.
>
> | Fine-grained dataset in [R1] | Q-Instruct | GPT-4V | EDQA (Ours) |
> |-----|-----|-----|-----|
> | CSIQ (Same type, various levels) | 0.115 / 0.081 / 0.557 | 0.419 / 0.402 / 0.906 | **0.955** / 0.925 / 0.958 |
> | CSIQ (Same level, various types) | 0.117 / 0.069 / 0.416 | 0.325 / 0.244 / 0.482 | **0.905** / 0.690 / 0.857 |
> | SPAQ (Various score regions) | 0.448 / 0.233 / 0.328 | 0.653 / 0.398 / 0.448 | **0.921** / 0.596 / 0.961 |
>
> Also, we provide the statistics of our model's confidence. The results below show that the confidence of consistent prediction is much higher than inconsistent prediction, which reflects the self-evaluation ability of our model.
>
> | Confidence statistics | consistency | inconsistency |
> |-----|-----|-----|
> | CSIQ (Same type, various levels) | 0.933 ± 0.120 | 0.629 ± 0.083 |
> | CSIQ (Same level, various types) | 0.860 ± 0.141 | 0.611 ± 0.098 |
> | SPAQ (Various score regions) | 0.900 ± 0.125 | 0.649 ± 0.110 |
>
> We have added the results into Table A11, A12 and L1170 (Sec. D.2).
>
> **Q3. The maximum resolution.**
>
> Thanks for pointing out this confusion. We have clarified this in L1002 (Sec. C) of the revised version. Actually, without the vision abstractor, the maximum resolution is limited to 672. However, with the vision abstractor, we can process images with much larger resolutions (up to  $2500 \times 2500$). In our experiments, the maximum image resolution is $1092 \times 1456$, thus the resolutions of all images are retained.
>
> **Q4. Results in real-world images and extreme conditions.**
>
> Thanks for your suggestion. In the revised version, we evaluate our model's ability to assess in-the-wild real images in Fig. A14. In Fig. A9, we also assess our model's performance on images with severe distortions. Our model performs well in such extreme cases.

---

> > ### Comment · Reviewer_5EhH · 2024-11-25
> >
> > Thank you for the authors' response. My concern has been adequately addressed, and I am pleased to recommend this paper for acceptance.

---

> > > ### Author Response · Authors · 2024-11-25
> > > **Thanks to Reviewer 5EhH**
> > >
> > > Thanks for recognizing our work.
> > >
> > > Your valuable comments greatly help us improve our manuscript and deepen our understanding of comparison order effects.

---

### Official Review · Reviewer_itdg · 2024-11-02

**Soundness:** 3
**Presentation:** 3
**Contribution:** 3
**Rating:** 8
**Confidence:** 5

**Summary:**

This paper proposes an Enhanced Descriptive image Quality Assessment method based on the VLM, and constructs a large-scale multimodal dataset. Compared with the existing VLMs-based IQA algorithms and the traditional score-based IQA algorithms, the authors adopt a general method to achieve multi-functional quality assessment, which can obtain brief and detailed responses. The authors conducted sufficient comparative experiments and ablation experiments to illustrate the superiority of the proposed method. From the experimental results, the proposed method is capable of coping with a variety of IQA tasks and can generate reasonable quality descriptions. This work is innovative and inspiring and the paper is well organized.

**Strengths:**

The authors have achieved multi-functional quality assessment through a general method, including quantitative indicators and qualitative language descriptions. The proposed method can adapt to a variety of practical application scenarios. The authors have achieved the purpose of multifunctional quality assessment based on VLM by reasonably designing data tags and prompts. The proposed multi-functional IQA task paradigm can provide support for subsequent work. The proposed model has achieved good performance on a variety of evaluation tasks, which shows its effectiveness.

**Weaknesses:**

As shown in the experimental results，the proposed method shows good multi-functional IQA performance. However, the model size of the proposed model is much larger than that of the traditional IQA models, and the model training requires more computing resources. In particular, for the classic quantitative quality assessment task, the cross-modal training adopted by the proposed model does not seem to achieve significant improvement compared with the SOTA single-modal IQA model.

**Questions:**

The reviewers want to know the training resource consumption of the proposed model, as well as the inference efficiency. In addition, whether cross-modal and multi-functional IQA tasks are beneficial to classic instant rating IQA task.

---

> ### Author Response · Authors · 2024-11-21
> **Responses to Reviewer itdg**
>
> Thanks for the reviewer's efforts and appreciation of our comprehensive task paradigm, effectiveness of dataset construction method, application potential, and strong performance. Below we address the main concerns.
>
> **Q1. Training and inference costs.**
>
>   Thanks for the suggestion. The training and inference costs are presented below and added in Sec. C.1.
>
>   - **Training costs.** The trainable parameters are 70M (54M for vision abstractor and 16M for LoRA), constituting only 0.98% of the total parameters (7.11B). The model is trained on 8 GPUs (RTX A6000). The training is completed in around 22 hours.
>
>   - **Inference costs.** The inference latency depends on the response length and it is tested on a single RTX A6000 GPU. For example, for brief tasks task with the short answer prompt (about 2.92 words), the inference time stands at approximately 2.23s / batch=32, transformed to 0.07s / sample. For the assessment reasoning task (75.84 words on average), the inference time is 22.97s / batch=32 (i.e., 0.72s per response). Despite the incorporation of resource-intensive MLLMs, EDQA remains deployable on a single consumer GPU (e.g., RTX3090).
>
> **Q2. Beneficial relationships between multi-modal task and instant rating task.**
>
> Thanks for the suggestion. We conduct experiments to study the relationships between multi-modal tasks (here we use comparison reasoning) and instant rating task.
>
> - Comparison reasoning task improves the performance on four instant rating datasets, but decreases the results on two datasets. Averagely, the comparison reasoning task helps the instant rating performance.
>
> - Instant rating task stably improves the performance on the comparison reasoning task.
>
> We have added these results to Table A10 and Sec. D.2.
>
> | Instant Rating Results | BAPPS      | KADID      | PIPAL      | TID2013    | LIVE-MD    | MDID2013   | Mean       |
> |-----|-----|-----|-----|-----|-----|-----|-----|
> | Only Rating            | 81.6 / 81.6 | 92.4 / 92.3 | 89.1 / 89.0 | 94.2 / 94.1 | **92.9 / 92.7** | **92.1 / 91.7** | 90.4 / 90.2 |
> | Co-training            | **84.7 / 82.4** | **93.6 / 93.1** | **90.5 / 90.0** | **96.9 / 96.4** | 92.1 / 91.8 | 90.0 / 89.6 | **91.3 / 90.6** |
>
> | Comparison Reasoning Results | | GPT-4 Score | BLEU | ROUGE-L |
> |-----|-----|-----|-----|-----|
> | Single-distortion | Only Reasoning | 74.3 / 69.6       | 0.203 / 0.202    | 0.465 / 0.453    |
> || Co-training | **75.1 / 74.9**   | **0.207 / 0.207** | **0.466 / 0.463** |
> | Multi-distortion | Only Reasoning | 70.6 / **69.1**   | 0.165 / 0.165    | 0.414 / 0.407    |
> || Co-training | **71.7** / 68.7   | **0.176 / 0.172** | **0.420 / 0.413** |

---

### Official Review · Reviewer_eDer · 2024-11-02

**Soundness:** 3
**Presentation:** 3
**Contribution:** 3
**Rating:** 6
**Confidence:** 5

**Summary:**

The authors propose a method called Enhanced Descriptive Image Quality Assessment (EDQA), which aims to leverage a large-scale multi-modal dataset (EDQA-495K) for improved IQA. The methodology includes both single-image assessments and paired-image comparisons, supported by an extensive distortion library. The paper claims significant performance improvements over existing models.

**Strengths:**

Strengths

1. The use of VLMs for IQA is new and aligns with current trends in AI research.
2. The authors propose a comprehensive and large-scale dataset, EDQA-495K, which could be a valuable resource for the research community.
3. The paper presents a clear and well-organized structure.

**Weaknesses:**

1. More detailed information is needed regarding the testing of the OOD settings presented in Table 3. Additionally, some non-reference IQA models, such as LIQE, are capable of identifying distortion types as well. It is recommended that these methods be included in the comparison.

2. Table 4 should incorporate IQA datasets that feature realistic distortions, such as KonIQ-10k and SPAQ. Including these datasets will allow for a more robust evaluation of the model's generalization capabilities.

3. The provided examples indicate that the model consistently generates template-like responses. It is important to investigate whether the model can effectively address open-ended questions. If so, results from the Q-Bench benchmark, which includes questions for both single and paired image inputs, should be provided.

4.  The methodology for computing quality score regression results in the full-reference setting (Table A3) is not clearly defined. Please elaborate on this aspect.

5.  Are there any specific fine-tuning procedures employed to obtain the results presented in Table A4?

6. Given that the model supports a maximum of two images as input, this limitation may lead to cumbersome and time-consuming comparisons when analyzing model-processed images across a larger variety of contents and algorithms. Suggestions for addressing this issue would be appreciated.

7. The results of Q-Instruct and Co-Instruct should also be included in Table 6.

8. Are there any inherent limitations associated with this model? A detailed discussion on this topic would enhance understanding.

**Questions:**

See weaknesses.

---

> ### Author Response · Authors · 2024-11-21
> **(Part 1) Responses to Reviewer eDer**
>
> We thank the reviewer for the reviewer's work and appreciation of our novelty, dataset contribution, and clear writing. Below we address the main concerns point-by-point.
>
> **Q1.1 Details of the OOD setting.**
>
> To achieve an OOD distortion identification, for one category of distortion (e.g., compression), we train the model on some sub-categories (JPEG compression), and then assess on other sub-categories (JPEG2000 compression). The detailed split of the training and validation distortions are given in Table A2 of the revised version.
>
> **Q1.2 Results of LIQE.**
>
> Thanks for the suggestion. We have added the results of LIQE in Table 3. LIQE achieves comparable results with Co-Instruct, falling behind our EDQA. Note that LIQE only accepts one image as input, and thus can only be assessed in the non-reference setting.
>
> || GPT-4V | Co-Instruct | LIQE | EDQA | EDQA (OOD) |
> |-----|-----|-----|-----|-----|-----|
> | Single-dist. | 45.2 | 34.4 | 33.1 | 94.1 | 73.2 |
> | Multi-dist. | 39.8 | 33.3 | 31.4 | 89.3 | 77.2 |
>
> **Q2. Results on KonIQ and SPAQ.**
>
> Thanks for the suggestion. We have provided the results on in-the-wild KonIQ and SPAQ datasets in **Q1 to reviewer 5EhH** and Table A6. The results demonstrate the effectiveness of our method.
>
> **Q3.1 Template-like responses.**
>
> The template-like descriptions come from the alignment to the formats of training samples. When constructing detailed responses, we include three aspects of information: (a) brief content description, (b) distortion analysis, and (c) overall judgment. After training, the model tends to predict the three aspects of information, forming template-like descriptions.
>
> In the early stages of VLM-based IQA methods, if the descriptions are fully flexible, defining desirable target descriptions will be extremely hard. Because of this, in the current version, we mainly focus on standardized answers, which help easily evaluate, analyze, and improve the model. More flexible responses could be an interesting future exploration.
>
> **Q3.2 Results on Q-Bench.**
>
> Q-Bench primarily consists of multiple choice questions, while our model emphasizes descriptive ability, creating a disparity for direct assessment. To demonstrate that our dataset can enhance performance on Q-Bench, we select another open-source dataset containing multiple choice questions, Q-Instruct. Co-training on Q-Instruct and our dataset consistently outperforms training solely on Q-Instruct, highlighting the benefits of our dataset. Additionally, the results of LLaVA trained on Q-Instruct, as reported in Q-Instruct, are provided for reference.
>
> | Q-Bench | Yes-or-No | What | How | Distortion | Other |
> |-----|-----|-----|-----|-----|-----|
> | LLaVA trained on Q-Instruct | 66.4 | 58.2 | 50.5 | 49.4 | 65.7 |
> | EDQA trained on Q-Instruct | 70.5 | 49.6 | 51.7 | 55.4 | 56.7 |
> | EDQA trained on Q-Instruct + data | 74.0 | 62.8 | 54.0 | 66.7 | 63.9 |
>
> **Q4.  Details of the method for full-reference score regression.**
>
> The full-reference score regression results are evaluated on the PIPAL, KADID, TID2013, and CSIQ datasets. These datasets include high-quality reference images and their distorted versions under various distortions. We calculate the win rate of an image against others to determine its quality score.
>
> Specifically, for one image, A, we randomly sample comparison candidates, such as B, C, D, etc., which share the same content as A but have different distortions.
>
> Image A is then compared pairwise with each of its comparison candidates (B, C, D, etc.). In the full-reference setting, three images including the reference image, Image A, and one compared candidate are together input into our model for comparison.
>
> Finally, the win rate of Image A against its compared candidates is calculated as its quality score.
>
> We have added these details in L1059 (Sec. D.1) of the revised version.
>
> **Q5. Specific fine-tuning procedures?**
>
> No. The model is trained with the commonly adopted next-token prediction loss on our collected datasets. Remind that Table A4 in the original submission becomes Table A5 in the revised version.
>
> **Q6. Suggestions to improve the comparison efficiency.**
>
> We mainly focus on accuracy in this work, but there are some ways to improve efficiency, like a dynamic Elo rating method.
>
> First, score initialization. Each image is initially assigned an equal score.
>
> Second, random pair comparison. Two images are randomly selected for comparison. The winner's score increases, and the loser's score decreases using the Elo rating system.
>
> Third, dynamic grouping. Images with similar scores are grouped based on a threshold, minimizing unnecessary comparisons between highly divergent pairs.
>
> Finally, intra-group comparisons. Comparisons are performed within each group, and scores are updated dynamically.
>
> The last two steps repeat until scores stabilize or a pre-defined number of iterations is reached. This method can narrow down top-performing images while reducing computational overhead.

---

> ### Author Response · Authors · 2024-11-21
> **(Part 2) Responses to Reviewer eDer**
>
> **Q7. Results of Q-Instruct and Co-Instruct with BLEU and ROUGE-L metrics.**
>
> Thanks for the suggestion. We have added the results of Q-Instruct and Co-Instruct with BLEU and ROUGE-L metrics in Table 6. The assessment reasoning results in the non-reference setting are shown in the table below. See Table 6 in the revised version for all results.
>
> || Single-distortion | Single-distortion | Multi-distortion | Multi-distortion |
> |-----|-----|-----|-----|-----|
> || BLEU | ROUGE-L | BLEU | ROUGE-L |
> | Q-Instruct | 0.003| 0.210 | 0.002| 0.198 |
> | Co-Instruct | 0.005| 0.204 | 0.003| 0.203 |
> | EDQA | 0.129| 0.422 | 0.170| 0.415 |
>
> **Q8. Limitations.**
>
> Thanks. We have discussed the limitations in Sec. 6. We also provide some possible solutions to address these limitations.
>
> First, the fine-grained abilities requiring more high-level perception skills are still unsatisfactory. One possible way is to take the segmentation model (e.g., SAM) to add various distortions to different regions.
>
> Second, for the convenience of evaluating, analyzing, and improving the model, we mainly focus on standardized answers. To achieve more flexible responses, LLM rewriting and human annotation can be introduced to increase linguistic diversity during dataset construction.
>
> Third, whether our assessment can be used as feedback to improve the quality of generation or restoration models is still under-explored. Existing image restoration models (e.g., SUPIR [R1]) have demonstrated that diffusion-based restoration models can adjust some low-level textures according to the textual inputs. Therefore, one possible application is to co-train a diffusion-based restoration model with both images and quality-assessed texts to enhance the restoration quality.
>
> [R1] Scaling Up to Excellence: Practicing Model Scaling for Photo-Realistic Image Restoration In the Wild, CVPR2024

---

### Official Review · Reviewer_BK5E · 2024-11-03

**Soundness:** 2
**Presentation:** 3
**Contribution:** 2
**Rating:** 3
**Confidence:** 5

**Summary:**

In this paper, a large-scale EDQA-495K dataset is proposed to fine-tune VLMs as multi-functional image quality assessors that can handle both quality assessment and comparison tasks. In specific, they use a ground-truth-informed approach and retain image resolution during training to construct the dataset. Experimental results demonstrate the superiority of DepictQA-Wild over traditional score-based methods, prior VLM-based IQA models, and GPT-4V across various tasks and settings.

**Strengths:**

The introduction of the DQ-495K dataset represents a significant contribution by providing a large-scale, high-quality dataset for Image Quality Assessment (IQA). The paper discusses various IQA tasks within both full-reference and no-reference frameworks, including assessments of single images and comparisons between paired images. Additionally, the incorporation of confidence estimation for responses is an interesting enhancement that could potentially improve the reliability of the assessments.

**Weaknesses:**

Although this paper contributes a comprehensive dataset and a multi-functional model for descriptive image quality assessment, the major weakness is the limited novelty and contribution, with some consideration that it is somewhat a data extension of the existing dataset and pipeline. Specifically, the pair-wise quality assessment and reasoning issue has been well addressed in the previous work Co-Instruct, while the performance gain of the proposed model basically comes from the augmented instruction data, which is somehow not a considerable contribution.  Besides, CoInstruct can also handle the quality comparison between multiple images, which is a harder task compared to image pairs, while the proposed model can only handle the latter one. Therefore, while the formulation of the augmented dataset is potentially useful, neither new perspectives nor new techniques are raised in this paper that can warrant a high score compared to previous work.

**Questions:**

See my comments above.

---

> ### Author Response · Authors · 2024-11-21
> **Responses to Reviewer BK5E**
>
> Thanks for the reviewer's efforts and appreciation of our dataset contribution, comprehensive task paradigm, and potential for confidence estimation. We address the main concerns below.
>
> **Reply to a wrong submission?**
>
> The reviewer referred to our method as "**DepictQA-Wild**", whereas our method is called "**EDQA**". Additionally, there are other details in the review that do not align with our paper. It seems possible that the reviewer may have responded to the wrong submission/paper. Could we kindly ask the reviewer to double-check if the comments are indeed intended for this submission?
>
> **Q1. Well solved in previous work?**
>
> The reviewer claims that the pair-wise quality comparison task has been well addressed in Co-Instruct, which contradicts the experimental results.
>
> - As shown in Table 4, the comparison ability of Co-Instruct (52.0) is still largely inferior to PSNR (80.2) and SSIM (78.8) in the full-reference setting. Also, in the non-reference setting, Co-Instruct (69.8) falls behind MUSIQ (76.9) and GPT-4V (75.2).
> - As given in Table 5, Co-Instruct's comparison reasoning ability (37.6 / 48.1) is also largely inferior to GPT-4V (66.2 / 60.3).
>
> These results prove that the pair-wise quality comparison task is still far from the reviewer-claimed "well solved".
>
> **Q2. Should a high-quality dataset be considered as a contribution?**
>
> The review claims that the instruction dataset "is somehow not a considerable contribution", which contradicts the literature on IQA. Most previous VLM-based IQA methods focus on dataset reconstruction, including Q-Instruct (Wu et al., CVPR2024), Co-Instruct (Wu et al., ECCV2024), and DepictQA (You et al., ECCV2024). This is because dataset quality is critical for VLM training, and the current VLM-based IQA datasets are still far from ideal and become a bottleneck in this area.
>
> Moreover, creating a high-quality dataset is not as trivial as simply combining more data. We need to carefully design the distortion library, the brief-detail combined dataset framework, and the prompt method. With our comprehensive distortion library and ground-truth-informed generation process, we successfully synthesize accurate training data.
>
> **Q3. Comparison number.**
>
> First, the key to the quality comparison task is the comparison accuracy, rather than the comparison number of one-time input. As stated in **Q1**, the pair-wise comparison ability of Co-Instruct is still largely inferior to PSNR and SSIM in the full-reference setting. Discussing comparison numbers regardless of comparison accuracy is unreasonable.
>
> Second, there are several ways to extend the pair-wise comparison to multi-image comparison, such as Elo rating, round robin, ranked pairs, and so on.

---

> > ### Comment · Reviewer_BK5E · 2024-11-26
> > **Declaration on the typo.**
> >
> > Thanks for the declaration from the authors.
> > The mentioned name is not the wrong paper, but an Arxiv version of this paper was posted several months ago.
> > You can simply regard this as a **typo**, because the core content of these two papers is highly similar, and the comments are based on the current submitted version.
> > Thanks.

---

> ### Author Response · Authors · 2024-11-26
> **Thanks to Reviewer BK5E and welcome to discussion**
>
> Dear Reviewer BK5E,
>
> Thank you once again for your time and valuable comments.
>
> We have provided detailed responses to your concerns and believe that these address all the issues you raised. Please let us know if any aspects of our work remain unclear. We understand your time is valuable, and we greatly appreciate it if you could review our responses at your earliest convenience.
>
> In particular, **regarding the discrepancies in method names and details that do not align with our submitted manuscript, we sincerely hope you clarify this misunderstanding.**
>
> Thank you for your kind consideration.
>
> Sincerely,
>
> Authors

---

> > ### Comment · Reviewer_BK5E · 2024-11-26
> >
> > I appreciate the author's highlighting of the contribution of this paper, and I agree that the dataset also matters.
> > However, a top-tier conference expects work that can dig deeper into the cause of the problems that exist in previous works instead of a SOTA-chaser with piles of data and skills. Therefore, I can only raise my score to 5 after reconsidering the contribution. Thanks.

---

> > > ### Author Response · Authors · 2024-11-26
> > > **Thanks to Reviewer BK5E**
> > >
> > > Thanks for confirming our contribution and sharing much more positive feedback.
> > >
> > > We sincerely appreciate the time and effort you put into reviewing our manuscript, which greatly helps us improve our manuscript.

---

### Author Response · Authors · 2024-11-24
**Explanation of manuscript revision and welcome to discussion**

We sincerely thank all reviewers for their time and appreciation of our work. We are delighted that the reviewers have recognized our dataset contribution, multi-functional task paradigm, confidence estimation, and clear writing. Here we provide a detailed explanation of our revisions (highlighted in blue in the revised version).

- Revision of the main paper

  - Table 3. We add the distortion identification results of LIQE. (Q1.2 of reviewer eDer)
  - Table 6. We add the reasoning results of Q-Instruct and Co-Instruct with BLEU and ROUGE-L metrics. (Q7 of reviewer eDer)
  - Limitations in Sec. 6. We discuss possible solutions to address the limitations. (Q8 of reviewer eDer)

- Revision of the supplementary appendix

  - Table A2 and L915. We detail the OOD setting of distortion identification. (Q1.1 of reviewer eDer)
  - L1002. We clarify that our model can retrain the resolution of all images in our experiments. (Q3 of reviewer 5EhH)
  - Sec. C.1. The training and inference costs are presented. (Q1 of reviewer itdg)
  - Table A6. We add score regression results on a new dataset, KonIQ. (Q2 of reviewer eDer, Q1 of reviewer 5EhH)
  - Table A10. We analyze the beneficial relationships between multi-modal task and instant rating task. (Q2 of reviewer itdg)
  - Table A11, A12 and L1170. The effects of input order are studied. (Q2 of  reviewer 5EhH)
  - Fig. A9. We provide more qualitative results on severe distortions. (Q4 of  reviewer 5EhH)

**We kindly request reviewers to review our revisions. If there are any remaining issues, we would be more than happy to discuss them further.**

---

### Meta-Review · Area_Chair_t7nw · 2024-12-26

**Metareview:**

This paper proposes a multi-functional VLM-based IQA method. The proposed EDQA model can handle both the image quality assessment and comparison tasks, and full-reference and non-reference scenarios. The dataset is implemented with 35 types of synthetic distortions.

However, there were concerns that the dataset did not incorporate natural distortions.
There lacks a discussion on the use of synthetic and natural distortions.
Also, adding more details in the literature survey on datasets for distortion recognition could elaborate on the contribution of this work better. Currently, only 2 works without the released distortion lables are mentioned.
In the added evaluation on SPAQ / KoniQ dataset in Table A6, EDQA-LoRA trained on SPAQ dataset is only marginally better than Q-Align-LoRA while Q-Align-LoRA trained on SPAQ outperforms EDQA-LoRA original by a larger gap.

Meanwhile, while the scale of the dataset is one of the major contribution, the details of scaling up are not very clear other than citing You et al., 2023.

Another major concern is the limited novelty. Compared with an earlier work, Co-Instruct, this work seems to be incremental although the results are better. Basically, the dataset could be summarized as using the images from KADIS-700K dataset and using ChatGPT to create candidate questions, applying synthetic distortions and the corresponding templated responses.
While the authors state in the discussion with BK5E that creating high-quality data is not as trivial as simply combining more data,
that seems to describe the generation process of the proposed dataset.

To summarize, this paper could be improved by adding more analysis and discussion on the employment of real and synthetic distortions, by adding more extensive literature survey, and by better justifying the novelty of the contribution.

**Additional Comments On Reviewer Discussion:**

The reviewers find the proposed dataset to be valuable in the community to promote research in the IQA field. The effectiveness of EDQA is recognized in the evaluation benchmarks.

The reviewers suggested several ways of supplementing the validation by testing on datasets with real distortions. In addition to providing experimental results, the authors could provide more analysis and discussion.

Template-like responses are another concern. While the authors leave it as future work, it still remains as an issue.

---

### Decision · Program_Chairs · 2025-01-22

Reject